

# Comparison of aerosol LIDAR retrieval methods for boundary layer height detection using ceilometer backscatter data

Vanessa Caicedo[1], Bernhard Rappenglueck[1], Barry Lefer[2], Gary Morris[3], Daniel Toledo[4], and Ruben Delgado[5]

[1]Department of Earth and Atmospheric Sciences, University of Houston, Houston, TX, USA

[2]Tropospheric Composition Program, Earth Science Division, NASA Headquarters,Washington, DC, USA

[3]School of Natural Sciences, Saint Edward's University,Austin, TX, USA

[4]Department of Physics, University of Oxford, Oxford, UK

[5]Joint Center for Earth Systems Technology, University of Maryland Baltimore Country, Baltimore, MD, USA

*Correspondence to:* Vanessa Caicedo (caicedo.vanessa@gmail.com)

**Abstract.** Three algorithms for estimating the boundary layer heights are assessed: an aerosol gradient method, a cluster analysis method, and a Haar wavelet method. Over 40 daytime radiosonde profiles are used to compare aerosol backscatter boundary layer heights retrieved by a Vaisala CL31 ceilometer. Overall good agreement between radiosonde and aerosol derived boundary layer heights was found for all methods. The cluster method was found to be particularly sensitive to noise
5 in ceilometer signals and lofted aerosol layers (48.8% of comparisons), while the gradient method showed limitations in low aerosol backscatter conditions. The Haar Wavelet method demonstrating to be the most robust only showing limitations (22.5% of all observations) due to the basic assumptions used to derive BLH from aerosol backscatter concentrations rather than errors with the algorithm itself. Disagreement between thermodynamically and aerosol derived boundary layer heights and the methodology used to estimate these heights was seen with all methods.

## 1 Introduction

10 The boundary layer (BL) is defined as the lowest layer in the atmosphere directly influenced by the earth's surface. The boundary layer reacts to surface forcings such as evaporation and transpiration, heat transfer, frictional drag, and terrain-produced air flows within a time scale of an hour or less (Stull, 1988). Other forcings such as pollutant emission in particular PM 2.5 (Particulate Matter) can enhance the stability of the BL and decrease the boundary layer height (Petäjä et al., 2016). 15 Above the boundary layer is the free troposphere (FT) acting as a cap to the BL. Convection and turbulence created by surface heating leads to the gradual growth of the BL starting at sunrise, mixing gaseous compounds and particles within the convective mixing layer (ML). Above the ML is the stable entrainment zone (EZ), where the FT is entrained downward into the ML, and ML thermals overshoot upward into the EZ (Stull, 1988; Toledo et al., 2014). The ML begins to decay as surface heating and turbulence decrease eventually creating a near surface nocturnal stable layer (NSL). Left over constituents from the daytime ML 20 form the residual layer (RL) above the NSL (Stull, 1988). More complex BL structures can also form in specific environmental conditions such as multiple stable layers and internal boundary layers (Garratt, 1990; Stull, 1988).



The determination of the BL height (BLH) is vital in air pollution studies as it determines the extent of vertical mixing of pollutants. While this is a key parameter in air pollution modeling and air quality studies, continuously monitoring of the BL is rarely available. The most common way of retrieving the BLH has been done with the use of radiosondes. However, radiosondes are seldom launched more than a few times a day except during extensive and costly scientific campaigns in which they are only

launched for the duration of the campaign. Apart from a few occasions (e.g., André and Mahrt, 1982; Berman et al., 1999; Day et al., 2010), NSL measurements are particularly uncommon since most radiosonde launches are performed during daytime hours. In recent years, remote sensing techniques such as Light Detection and Ranging (LIDAR), Radio Acoustic Sounding Systems (RASS) and Sonic Detection and Ranging (SODAR) systems have allowed for the continuous monitoring of the BL (Cohn and Angevine, 2000; Seibert et al., 2000; Emeis et al., 2004, 2006; Eresmaa et al., 2006; Baars et al., 2008; McKendry

et al., 2009; Muñoz and Undurraga, 2010; Emeis et al., 2012; Haman et al., 2012; Compton et al., 2013; Scarino et al., 2014; Uzan et al., 2016). Ceilometers in particular offer a low maintenance and low cost solution to constantly monitoring the BLH using backscatter from aerosol particles, facilitating the monitoring of the nocturnal stable layer, internal aerosol layers and the nighttime residual layer (Haman et al., 2012, 2014; Pandolfi et al., 2013; Peña et al., 2013). The extensive data set from continuous LIDAR measurements results in the need for the determining the most reliable and accurate method to be used in

automated retrievals.

In order to evaluate the retrieval of BLHs from aerosol LIDARS, we tested three distinct methods. This study will evaluate a gradient method, a Haar Wavelet method , and a Cluster Analysis method to retrieve BLH using aerosol backscatter measured by a Vaisala CL31 ceilometer located in an urban environment. These BLHs are then compared to radiosonde derived BLHs for validation. The effect of cloud signals on the retrieval of the BLH is also observed in all retrieval methods tested and discussed

in this study.

## 2 Data and Instrumentation

This study uses Vaisala CL31 ceilometer data and radiosonde profiles measured at the University of Houston (UH) Main Campus. UH Main Campus is located about 70 km northwest of the Gulf of Mexico and 5 km southeast of downtown Houston. The UH CL31 was mounted a top a trailer approximately 3.5 m above ground and radiosonde launches were performed next

to the CL31 trailer. Over 80 radiosonde launches and CL31 backscatter profiles beginning in January 2011 through March 2015 are used. Only cloud-free launches are used in the BLH comparison (total of 48 launches). The effect of cloud signals are analyzed separately for each method in Section 4.4. In addition, this data set includes data from the NASA DISCOVER-AQ (Deriving Information on Surface conditions from Column and Vertically Resolved Observations Relevant to Air Quality) Texas campaign in September 2013.

### 2.1 Vaisala CL31

The Vaisala CL31 ceilometer operates at a wavelength of 905 nanometers (nm) using an indium gallium arsenide laser diode (InGasAs) system with a 1.2 microjoule (mJ) pulse for 110 nanoseconds (ns) and mean pulse repetition rate of 8192 Hertz



(Hz). It uses a single lens design to both transmit and receive light signals. This design reduces the optical crosstalk between transmitter and receiver and in turn increases the signal-to-noise ratio. A beam splitter gives full overlap of the transmitter and receiver field-of-view at an altitude of 0m (Münkel et al., 2007).

The backscatter coefficient $\beta(x, \lambda)$ or the scattering cross section per unit volume can be calculated with the following formula:

$$P(x,\lambda) = \frac{c}{2x^2} P_0 A \eta O(x) \Delta t \times \beta(x, \lambda) \tau^2(x, \lambda), \tag{1}$$

where $P$ is the optical power received by the ceilometer from distance $x$, $c$ is the speed of light, $\Delta t$ is the pulse duration, $P_0$ is the average laser power during pulse, $A$ is the area of receiver optics, $\eta$ is the receiver optics' efficiency, $O(x)$ is the range dependent overlap integral between transmitted beam and received, $\tau(x, \lambda)$ is the transmittance of the atmosphere between LIDAR and volume, $\lambda$ is the wavelength of the emitted laser pulse, and $x$ is the distance between LIDAR and scattering volume. Backscatter profiles with signals from clouds, rain, or fog are identified as signals higher than $2000 \times 10^{-9} m^{-1} sr^{-1}$ and were not used for this BLH comparison (Kamp and McKendry, 2010).

The CL31 can measure aerosol backscatter up to 7500m. However, the CL31 does not record these signals, but instead only accumulates backscatter intensity every 16 seconds with a maximum height of 4500m and 10m resolution. For more in depth information about the instrument see Münkel et al. (2007).

## 2.2 iMet Radiosondes

Radiosondes launched at UH Main Campus are International Met Systems Incorporated model iMet-1. All radiosondes return GPS (Global Positioning System) location, GPS altitude, wind speed and direction, pressure, temperature, and relative humidity with a 1 Hz sampling rate using a 403 MHz transmitter. Radiosonde had a response time of 1s. Radiosondes used here have a resolution of 0.01hPa, a response time of 1s, and an accuracy of 0.5hPa for pressure measurements. Temperature sensing has a resolution of 0.01 °C , accuracy 0.2 °C, and a response time of 2s. The humidity sensors for the radiosondes have a resolution of less than 0.1%, accuracy of 5%, and a response time 2s. Average ascent rate for all launches was about 5 m/s.

A total of 85 launches were analyzed for this study, but only launches corresponding to cloud-free aerosol backscatter vertical profiles are used. A resulting 48 launches between January 2011 and March 2015 are used. All launches occurred between 6:00 and 17:00 CST. Only four launches happened before 9:00 with most launches happening around 13:00 CST.

## 3 Boundary Layer Height Retrieval Methods

All aerosol derived BLH methods presented here are based on two assumptions: 1) the BL contains constant concentration of aerosols therefore a constant concentration of backscatter due to convective and turbulent mixing and 2) the clean FT above will create a negative gradient in backscatter from higher concentrations within the BL towards lower concentrations in the FT. The local maximum of this gradient is identified as the top of the BL (Steyn et al., 1999). Thermodynamic radiosonde BLHs





are calculated using a skew-*T log-P* diagram method and are compared to aerosol derived BLHs within 10 minutes before or after radiosonde launch.

### 3.1 Skew-*T log-P* Diagram for Radiosonde Boundary Layer Heights

A stable BL is characterized by having an environmental lapse rate greater than a moist/dry adiabatic lapse rate (Fig. 1a), while
an unstable boundary layer is identified a dry adiabatic lapse rate greater than the environmental lapse rate (Fig. 1b). Stable profiles BLHs are identified as the top of the shallow stable layer as seen as a strong positive vertical gradient change in temperature and a strong negative gradient in dew point temperature profiles (Fig. 1a). BLHs during unstable conditions are identified as the base of the stable EZ (i.e. temperature inversion) where temperature profile intersects dry adiabates and/or where relative humidity or dew point temperature profiles sharply decreases as seen in skew-*T log-P* diagram Fig. 1b (Stull, 1988; Kovalev
and Eichinger, 2004; Haman et al., 2012). A previous study by Haman et al. (2012) found a correlation coefficient of 0.96 during unstable conditions and 0.91 during stable conditions when comparing ceilometer and radiosonde derived BLHs using the skew-*T log-P* method.

### 3.2 Vaisala Corporation Aerosol Backscatter Gradient

The Vaisala Corp. BL Matlab *v*1.3 algorithm is used in this study. This algorithm finds negative gradients with increasing
altitude in aerosol backscatter profiles following the assumptions discussed in Sect. 3. A 10 $\mathrm{minute}$ and 120 $\mathrm{meter}$ height averaging is applied to the profile along with a temperature dependence curve of -10 as recommended by Vaisala Corporation (C. Münkel, pers. comm., September 2013) due to the tendency of the CL31 having a curvature in backscatter profiles with increasing internal temperatures. The temperature correction of -10 adjusts the shape and curve of temperature affected backscatter profiles with negligible effects on aerosol layer detection (Münkel et al., 2007; Vaisala Oyj, 2011, C. Münkel, pers.
comm., April 2016).

The change in backscatter by height ($d\beta/dx$) is calculated by the algorithm, which then finds the largest three negative gradients with minimum backscatter of $200 \times 10^{-9} m^{-1} sr^{-1}$ . The largest of the negative gradients is usually defined as the BL (Münkel et al., 2007; Vaisala Oyj, 2011). However, the largest negative gradient does not always correspond to the BL (see results Sect. 4). Therefore, a manual analysis of the algorithm's three resulting layers (Fig. 2) is required in order to prevent the
incorrect identification of other aerosol layers. The algorithm gives three maximum negative gradients every 1-minute, these are averaged to 10 minutes for radiosonde comparison. The manual approach required to select one of the three maximum negative gradients as the BLH requires a priori knowledge of typical nocturnal and daytime BL heights. In addition, this manual selection analysis can be time-consuming especially when longterm LIDAR data is evaluated.

### 3.3 Cluster Analysis

This method uses variations in the measured aerosol vertical profiles for BLH calculations. The BLH is typically identified as the variance local maximum based on the assumption that the EZ contains high aerosol variability due to clean air masses





from the free atmosphere mixing with masses from the BL. The center of the EZ corresponds to the top of the BL (Hooper and Eloranta, 1986; Stull, 1988; Piironen and Eloranta, 1995).

Toledo et al. (2014) tested nonhierarchical and hierarchical cluster analysis on LIDAR retrieved vertical aerosol distribution and its variance. Both cluster methods were found to be reliable in calculating BLHs but with a tendency to overestimate the BLH compared to backscatter gradient methods. This overestimation was attributed to the gradient methods identifying the BLH as a significant decrease in signal, while the cluster method uses a local maximum in variance corresponding to the middle of the EZ. The maximum negative gradient does not always correspond to the local maximum in variance, in these cases the greater the EZ depth the greater the overestimation the BLH (Toledo et al., 2014). Nevertheless, the cluster method offers a unique BLH, whereas aerosol gradient methods can give multiple results

### 3.3.1 Data Processing for Cluster Analysis and Application

Due to low signal-to-noise ratio and noise-generated artifacts, both a 10-minute moving time average and moving height average was applied to raw backscatter profiles. Height averages were applied as seen in Table 1. These averaging settings were chosen as they created the most reliable cluster calculated BLHs, similar to findings in averaging done for gradient methods (Emeis et al., 2008a, b). Due to increasing noise in backscatter profiles with height, lower averaging was applied to lower altitudes while higher averaging was applied to higher altitudes (Table 1). This study found that these averaging settings worked best on most aerosol profiles and aerosol conditions. Typically, lower averaging than those listed in Table 1 caused artificial variance peaks, while greater averaging smoothed out variance peaks in the backscatter profiles. The moving average also leads to more profiles containing cloud signals; therefore only 45 comparisons were found to be valid for this method.

Variance $V$ as a function of height $z$ were then calculated from cloud-free profiles $P$ using the following formula (Hooper and Eloranta, 1986):

$$V(z) = \frac{1}{N-1} \sum_{i=1}^{N} [P(z,t_i) - \bar{P}(z)^2], \quad (2)$$

where $P(z,t_i)$ is the averaged LIDAR backscattered signal at time $t_i$ and height $z$, and $\bar{P}$ averaged profile from $N$ number of profiles.

K-means clustering can then be applied to identify BLHs. K-means is a data-partitioning algorithm that assigns observations to exactly one of k clusters defined by centroids (cluster centers), where k is chosen before the algorithm starts (Anderberg, 1973; Toledo et al., 2014). The algorithm works as follows:

Step 1. Choose k initial cluster centers (centroid).

Step 2. Compute point-to-cluster-centroid Euclidean distances of all observations.

Step 3. Assign each observation to the cluster with the closest centroid.

Step 4. Compute the average of the observations in each cluster to obtain new centroid locations

Step 5. Repeat steps 2 through 4 until cluster assignments do not change, or the maximum number of iterations is reached, whatsoever occurs first, depending on computational resources (Toledo et al., 2014).





Previous determination of the number of clusters present or needed in the dataset is required for cluster validation, since the number of clusters is a parameter to be introduced into the cluster algorithm (Step 1).

Cluster analysis will typically divide a well-mixed boundary layer into two clusters, one below a peak in variance corresponding the center of the EZ, and one above the variance peak (Fig. 3), however profiles with increasing noise and/or lofted

aerosol layers will cause the cluster analysis to assign clusters using other criteria (see results Section 4). The maximum height of these clusters are limited by the time of day to prevent the detection of other aerosol layers such as the top of the residual layer during nocturnal hours when only the NSL is of interest. Here, the maximum height for nighttime BL detection is 400m, whereas it is 2800m for daytime BL heights.

### 3.4   Haar Wavelet Method

Aerosol backscatter BLHs are derived with a Covariance Wavelet Transform utilizing the Haar wavelet compound step function with multiple user defined wavelet dilations (Cohn and Angevine, 2000; Davis et al., 2000; Brooks, 2003; Baars et al., 2008; Compton et al., 2013; Uzan et al., 2016). This method identifies the sharp aerosol backscatter gradient corresponding to the top of the BL by calculating wavelet transform at various dilations. The Haar wavelet function $h$ is defined as follows:

$$h(\tfrac{z-b}{a}) = \left\{ \begin{array}{ll} -1: & b - \tfrac{a}{2} \leq z < b \\ +1: & b \leq z \leq b + \tfrac{a}{2} \\ 0: & elsewhere \end{array} \right\}, \tag{3}$$

where $z$ is the vertical altitude in this application, $a$ is the vertical extent or dilation of the Haar function, and $b$ is the center of the Haar wavelet function. The covariance transform of the Haar wavelet function, $w_f$, is defined as:

$$w_f(a,b) = a^{-1} \int_{z_b}^{z_t} f(z) h(\tfrac{z-b}{a}) dz, \tag{4}$$

where $z_t$ and $z_b$ are the top and bottom altitudes in the backscatter profile, $f(z)$ is the backscatter profile as a function of altitude, and $a$ is the normalization factor or the inverse of the dilation, respectively.

Defining the dilation factor $a$ and center $b$ of the Haar wavelet function are key in correctly identifying the BLH using backscatter profiles. In this study, $b$ ranges from the lowest ceilometer recorded backscatter altitude of 10m to a maximum BLH of 2800m. This limit was set as no radiosonde derived BLHs were found above 2800m.

As with previous studies (Brooks, 2003; Baars et al., 2008; Compton et al., 2013; Scarino et al., 2014), the dilation factor $a$ affects the number of covariance wavelet transform coefficients (CWTC) local minimums. Lower dilation values create

numerous CWTC local minimums (Fig. 4b) at heights of smaller aerosol gradients in the measured profiles. Larger values create large local minimums (Fig. 4c and 4d) at the height of the biggest aerosol gradient in the backscatter profile (Fig. 4a). Here we use a dilation value of 30m for nighttime BLHs since the NSL tends to have a smaller backscatter gradient than the above RL creating a need for more than one local minimum (not shown). In these cases, the CWTC local minimum closest to the surface is chosen as the BL. A higher value of 300m (Fig. 4d) for the dilution factor $a$ is applied for daytime BLHs

to identify the sharp transition between ML and FT and to decrease signals from smaller aerosol gradients below the BLH.



Cloud-free CL31 backscatter profiles are averaged first vertically according to Table 1 followed by a 10-minute average before applying the Haar Wavelet algorithm.

## 4 Results

BLH retrieval methods are evaluated and quantified against radiosonde derived BLHs using bias and standard deviation calculated in accordance to Nielsen-Gammon et al. (2008) and Haman et al. (2012). Here, the bias is the difference between the means of aerosol retrieved BLH and the corresponding radiosonde BLH, and the standard deviation is the root-mean-square value of the departures of the individual pair sample differences from the bias. A two-sided, paired sample t test is used to define the statistical significance of the bias:

$$t = \frac{\bar{X} - \mu}{S} \sqrt{N},$$ (5)

where $\bar{X}$ is the mean aerosol BLH samples, $\mu$ is the radiosonde BLHs mean, $S$ is standard deviation of samples, and $N$ is the number of pair samples.

The null hypothesis is defined as unbiased aerosol derived BLHs when compared to radiosonde BLHs. It was not rejected when the calculated t-test value (t) was within $\pm 1.96$ and the p value was greater than 0.05 or 95% confidence, in alignment with previous approaches (Nielsen-Gammon et al., 2008; Haman et al., 2012).

### 4.1 Aerosol Backscatter Gradient Method Results

A previous study done by Haman et al. (2012) found that ceilometer BLHs derived from the aerosol backscatter gradient showed excellent correlation to radiosonde BLHs for both stable and unstable conditions, over a period of two years using more than 60 daytime radiosonde profiles. Haman et al. (2012) found the aerosol backscatter gradient capable of continuously identifying the height of the BL after manually choosing one of the three resulting aerosol layers, with limited detection following precipitation, and during periods of high wind speeds. Low aerosol content after rain events through wet deposition of aerosols and dispersion of aerosol due to high winds speeds limit the formation of aerosol layers, therefore limiting the detection of the BLH with aerosol gradients. These limitations however, are less relevant for air quality studies since typically these situations are also accompanied by lower pollutant levels (e.g. through air mass change, enhanced vertical mixing, enhanced dry deposition due to high winds, and wet removal of soluble gases during the preceding precipitation). Late afternoon hours also present a challenge since the discontinuous transition from unstable (ML) to stable boundary layer (NSL) can create multiple aerosol layers (Seibert et al., 2000; Haman et al., 2012). This is still an important time period for primary pollutant concentrations as they would still be critically determined by the BLH (in particular during evening rush hour), however the diurnal peak in photochemistry activity for build-up of secondary pollutants has passed making this a less crucial time for these pollutants.

This study found similar results using 47 cloud-free radiosondes. A correlation coefficient (r²) of 0.85 was found (Fig. 5) when comparing the aerosol backscatter gradient BLHs to daytime radiosonde BLHs. A bias of -42.5m and a standard





deviations of 209.5m (Table 2) were found (not statistically significant; $p < 0.05$). The bias indicates aerosol gradient method BLHs are generally lower than radiosonde BLHs. The overall agreement shows the ability of this method to calculate the BLH reasonably well once one of the three calculated aerosol backscatter gradients is chosen as the BL. However, this requires *a priori* knowledge of typical BLHs at the measurement site and a manual inspection of aerosol gradients calculated. In addition,

limited detection of the BLH was also seen in conditions with low aerosol content when the algorithm did not find small gradients in the backscatter profile. No combination of available setting options was found to improve BLH detection in these conditions. Furthermore, disagreement was found when the largest gradient in an aerosol profile does not correspond to the thermodynamic BLH found using radiosonde profiles. This is due to the assumption in the methodology of using aerosol gradients to detect the BLHs and thermal parameters to detect radiosonde BLHs.

## 4.2    Cluster Method Results

CL31 BLHs using the cluster method showed a slightly lower correlation than the aerosol gradient method with a correlation coefficient of 0.82 (Fig. 5), a bias of -61.0m and a standard deviation of 243.5m (Table 2). Disagreements found between radiosonde and cluster derived BLHs were most commonly due to noise in backscatter profiles and lofted aerosol layers. From the 45 comparisons performed, 13.3% showed the algorithm mistakenly finding maximum variance peaks not corresponding

to the BL but to noise or other aerosol layers (Fig. 6). Sixteen cases (35.5%) were found where noise created multiple variance peaks therefore the cluster analysis divided backscatter profiles into clusters of similar variance intensity (Fig. 6) rather than above and below a strong variance peak (Fig. 3). This division underestimated the BLH (bias of -61.0) since the cluster was divided into relatively low variance closer to the surface and high variance in higher altitudes. This is due to the fact that CL31 displays increasing noise with increasing altitude. Five instances were found where the variance maximum did not equal

radiosonde derived BLH due to signals from lofted aerosol layers. In these cases a smaller maximum corresponded to the BL. These were not algorithm erros but instead due to the implicit assumptions in using aerosol backscatter for BLH detection (constant concentration of backscatter within the BL and a negative gradient in backscatter corresponds to the top of the BL)

### 4.3    Wavelet Method Results

The Haar wavelet method showed excellent agreement when compared to 48 radiosonde BLHs with a correlation coefficient

of 0.89 (Fig. 5). Statistical analysis showed a bias of 51.1m (not statistically significant) and a standard deviation of 187.0m (Table 2). Disagreement was found when aerosol backscatter profiles contained multiple sharp gradients corresponding to lofted aerosol layers ($\sim 12.5\%$ of total cases). These shallow aerosol layers often have stronger gradients than that of the BL. In these cases, the second largest gradient is very often the BL ($\sim 67\%$). In addition, another $\sim 10\%$ of total cases showed deviations where the radiosonde derived BLH did not correspond to the greatest gradient in the aerosol profile as shown in

Figure 7. This disagreement and positive bias found can be attributed to the differences in determining BLHs using aerosols and thermodynamically using radiosondes. Aerosols can penetrate into the stable layer transporting aerosols to higher altitudes than the BLH (inversion height) causing an overestimation of aerosol derived BLHs (McElroy and Smith, 1991; Seibert et al., 2000). Removing the  22.5% of deviations falling into the cases described above would improve the correlation drastically ($r^2$





= 0.98). This provides confidence that all potential causes for deviations were identified. Overall, the wavelet method showed the best correlation of all methods when compared to radiosondes. In particular, this method was superior in the detection of BLHs in profiles with low aerosol backscatter. Under these conditions it was able to resolve weaker local maximums thus reasonably capturing the BLH.

## 4.4  BLH Retrieval with cloud signals

The identification of the BLH is particularly difficult in the presence of clouds when the daytime convective mixed layer is topped by cumulus or stratocumulus clouds. Strong cloud signals ($>2000 \times 10^{-9} m^{-1} sr^{-1}$) can limit the detection of the BLH due to the extinction of the backscatter signals above cloud layers. The effect of these cloud signals is observed for all BLH retrieval methods. Although this study observes daytime cloud signals, continuous ceilometer measurements may find similar signals during nighttime hours therefore our findings are not limited to daytime convective mixed layers. As an example, Figure 8 shows hourly aerosol backscatter profiles for September 15, 2013 and corresponding BLHs retrieved by the aerosol gradient, cluster and wavelet methods. Both aerosol gradient and wavelet methods consistently identify the BLH as the top of the cloud layer due to the large negative gradient created by strong cloud signals. This is often the height of the thermodynamic BL identified using relative humidity and dew point temperature methods, which find the height of the ML as the sharp decrease in moisture at the top of the cloud layer. Low cloud layers however impede the detection of the above BLH therefore the aerosol gradient and wavelet method will mistakenly identify the large gradient of the low cloud layers as the BLH, while the cluster method will identify the BL as the base of the low cloud layer. The aerosol gradient method typically found the BLH at the beginning of the large negative gradient (top of the cloud layer) while the wavelet method calculated the BLH slightly higher than the aerosol gradient method. Differences between these two methods were found to not exceed 200m and could be attributed to the different averaging settings applied for these methods.

The cluster method was found to constantly identify the cloud base as the BLH by assigning aerosol signals into a cluster of cloud signals and a second cluster of cloud-free signals with the first transition (BLH) of these clusters located at the base of the cloud layer (Fig. 9) at 970m. A second transition of clusters is located at the top of the cloud layer (about 1400m) corresponding to the BLHs retrieved by the aerosol gradient and wavelet methods. The cluster method then essentially calculates the cloud layer depth by assigning a cluster solely to the cloud layer.

The presence of clouds limits the detection of the BLH for all methods due to either the extinction of aerosol backscatter signals above the cloud, or the presence of low clouds mistakenly identified as the BLH. Hence the removal of profiles with cloud signals is needed for the automatic retrieval of the BLH. This affects the cluster and aerosol gradient methods in particular since the moving averaging performed before the application of the algorithms will expand cloud signals to a greater number of profiles subsequently eliminating these profiles for BLH detection.



## 5 Summary and Conclusions

Aerosol backscatter derived boundary layer heights from three distinct methods were tested and compared to radiosonde retrieved BLHs. An aerosol gradient method, a cluster analysis method, and a Haar wavelet method were compared to over 40 daytime radiosonde profiles using measured aerosol backscatter from a Vaisala CL31 ceilometer. The first method, the Vaisala Corp. aerosol gradient method finds the three largest gradients in a backscatter profile, one of which must be chosen as the height of the boundary layer. The second method, a cluster analysis method calculates variance in an aerosol backscatter profile with the BLH correlating to a peak in variance. K-means cluster analysis then divides a variance profile at the height of the BL (variance peak). The final method uses a Covariance Wavelet Transform utilizing the Haar wavelet compound step function to identify a sharp aerosol backscatter gradient corresponding to the top of the BL by calculating wavelet transform at various dilations. The results presented here used daytime measurements only, however the findings can be applied to similar signals as those found in the nighttime residual and nocturnal stable layers.

Overall good agreement was found for all methods with no statistically significant bias found. Yet all methods found cases where thermodynamic BLHs from radiosondes did not correlate with a maximum gradient in aerosol backscatter due to differences in thermodynamic and aerosol BLHs and the methodology used to derive these heights. The comparison between the aerosol gradient method and radiosonde derived BLHs showed difficulties in determining the BLH in low aerosol backscatter conditions. The calculation of the three largest gradients particular to this method was useful in situations where the largest gradient does not correlate with the radiosonde derived BLH. Yet this requires *a priori* knowledge of typical boundary layer heights and evolution in the location of interest. In contrast, the cluster method showed drawbacks due to sensitivity to noise generated artifacts or lofted aerosol layers where the algorithm mistakenly found peaks in variance and incorrectly identified them as the BLH. Profiles were also mistakenly divided into relative variance concentrations rather than a peak in variance, underestimating the height of the BL. Previous knowledge of the BL will aid in identifying these algorithm errors, but is not necessary to obtain a BLH with this method. Further work is needed to improve the cluster method sensitivity to noise.

The wavelet method showed the best agreement of all methods tested here, with 77.5% of cases showing excellent agreement with radiosonde BLHs without previous knowledge of the BL required to derive at a BLH. The cases where deviations occurred ($\sim$ 22.5% of all observations) were due to multiple sharp gradients corresponding to lofted aerosol layers and to the thermodynamically derived BLH not corresponding to the greatest gradient in an aerosol profile (Fig. 7). A bias of 51.1m was found indicating that wavelet method BLHs are generally higher than radiosonde derived BLHs. This disparity has been previously attributed to aerosol penetrating into the stable layer above the BLH leading to the overestimation of aerosol derived BLHs (McElroy and Smith, 1991; Seibert et al., 2000). The wavelet method also showed a higher ability of calculating the BLH under low aerosol conditions.

The effect of cloud signals in the determination of the BLH showed a clear difference between the negative gradient methods (aerosol backscatter and wavelet methods) and the cluster analysis method. Both aerosol gradient and wavelet methods identify the BLH as the top of the cloud layer where a sharp negative gradient created by strong cloud signals was found, while the cluster method identified the BLH as the base of the cloud layer. The cluster method was found to assign a cluster for cloud





signals and a cluster for cloud-free signals along a backscatter profile (Fig. 9). The automatic detection of the first transition of clusters identifies the BLH as the base of the cloud layer with the second transition at the top of the cloud layer, i.e. it identifies the cloud layer depth. Limited detection of the BLH in aerosol profiles with cloud signals is seen for all methods due to either the extinction of aerosol backscatter signals above the cloud, or the presence of low clouds mistakenly identified as

5  the BLH with the cluster and aerosol gradient methods being more sensitive due to the moving averaging applied expanding cloud signals to a greater number of profiles, consequently eliminating these profiles for BLH detection.

The results presented here demonstrate the ability of the Haar Wavelet method to more accurately detect BLHs than the aerosol gradient and cluster methods while requiring the least amount of manual inspection. The errors found with this method were due to the basic assumptions used to derive BLH from aerosol backscatter concentrations rather than errors with the

10  algorithm itself. For this reason, the use of this automated method is recommended for BLH observations with careful determination of the dilation values needed for individual instrument and locations. It is suggested to employ the wavelet method in future studies, in particular for long-term boundary layer evolution studies and spatial analysis of the BL using multiple LIDAR aerosol backscatter measurements. A combination of the wavelet method BLH retrievals during clear skies and the cluster analysis method's ability to calculate cloud depth is also recommended for more robust BL studies to retrieve more

15  information about the boundary layer under both conditions.

*Acknowledgements.* We wish to thank Christoph Münkel with all assistance provided with the ceilometer and Vaisala BL Matlab algorithm. James Flynn and Sergio Alvarez for assistance in the installation and maintenance of the ceilometer. Part of this work was funded by the Texas Commission of Environmental Quality (TCEQ) and the NASA DISCOVER-AQ project.



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





**Table 1.** Averaging heights by height range used on aerosol backscatter profiles for cluster and wavelet methods.

| Altitude Range | Averaging Height |
|:---:|:---:|
| 10-490 m | 70 m |
| 500-990 m | 330 m |
| 1000-1990 m | 590 m |
| 2000-4500 m | 690 m |



**Table 2.** Bias, Standard Deviation and number of data points (No.) for comparison of BLH retrieval methods to radiosonde BLHs.

| BLH Retrieval Method | Bias (m) | Standard Deviation (m) | No. |
|---|---|---|---|
| Aerosol Gradient | -42.5 | 209.5 | 47 |
| Cluster | -61.0 | 243.5 | 45 |
| Wavelet | 51.1 | 187.0 | 48 |





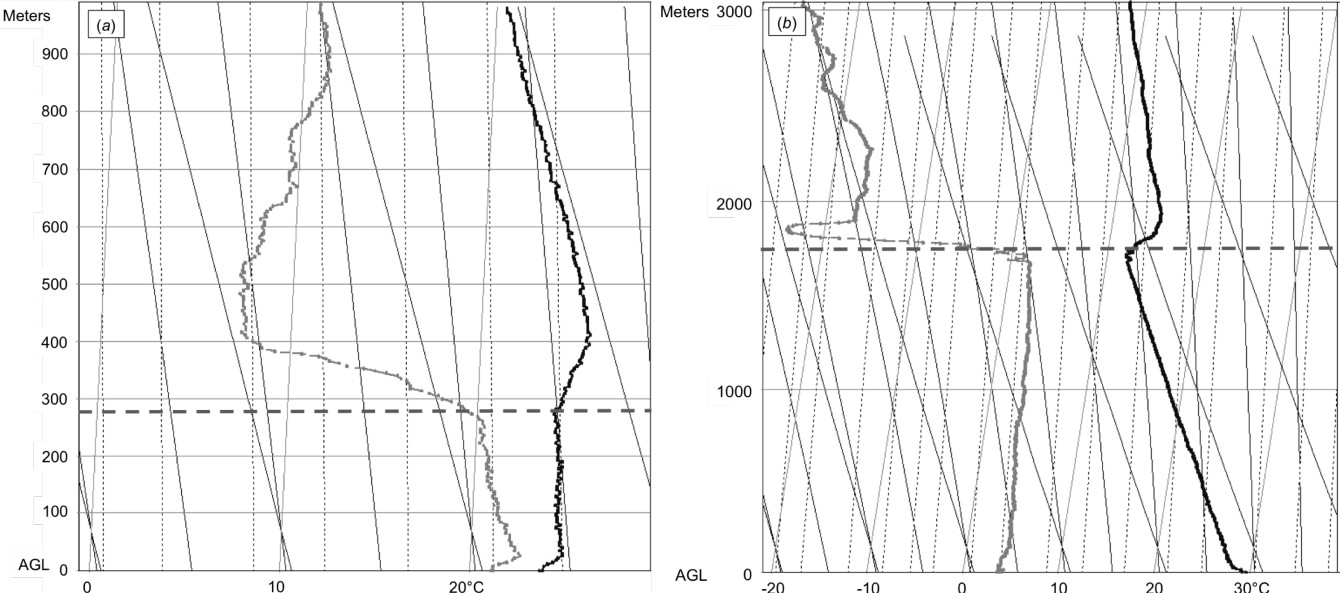

**Figure 1.** Skew-*T log-P* method for BLH detection using temperature (black) and dew point temperature (grey) for (a) stable and (b) unstable conditions with BLH shown as grey dashed line. Soundings from September 26, 2013 at 6:10 CST (a) and May 4, 2014 at 15:40 CST (b).





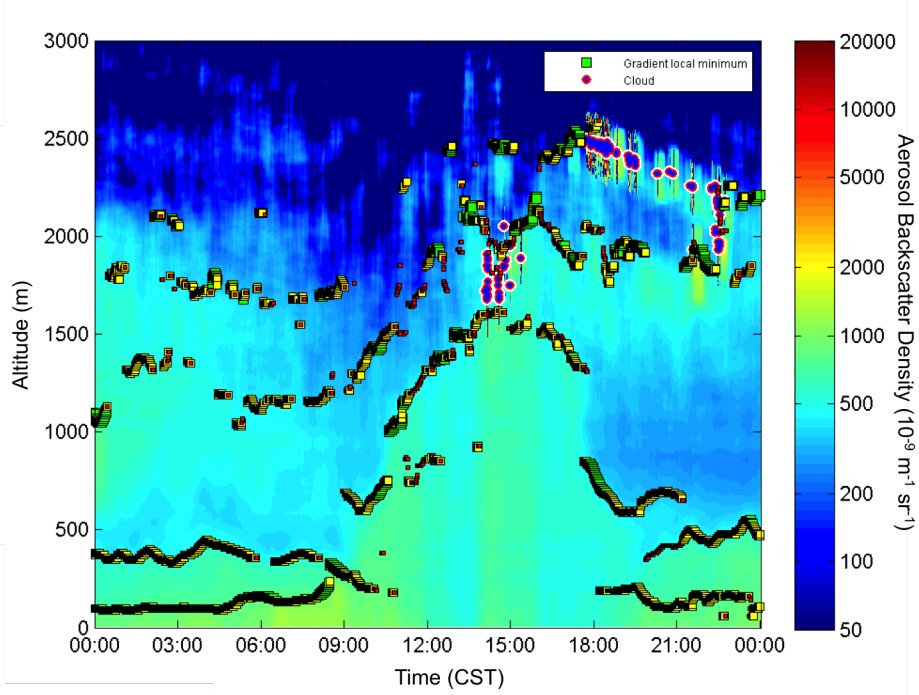

**Figure 2.** Aerosol backscatter density plot for September 26, 2013. Three gradient local minimums are plotted for each backscatter profile.





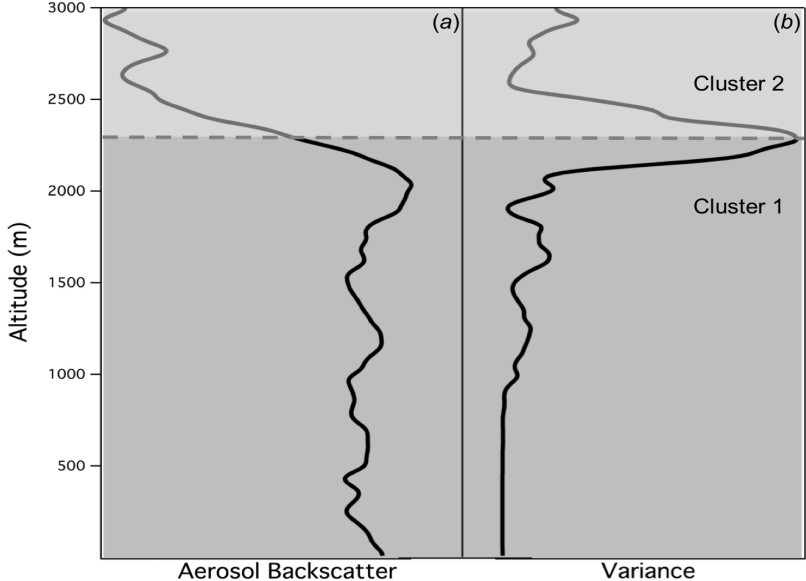

**Figure 3.** CL31 aerosol backscatter profile (a) and corresponding calculated variance profile (b) for September 25, 2014 at 14:30 CST. Dashed line shows the cluster derived BLH (2360m) at the height where the variance cluster assignment changes from cluster 1 to cluster 2.





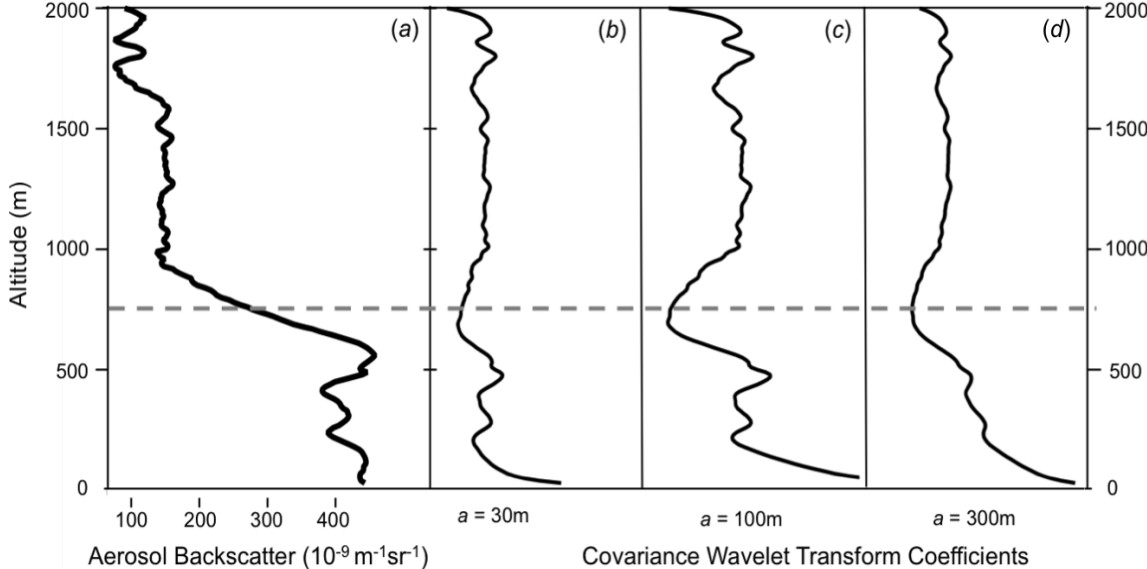

**Figure 4.** Daytime aerosol backscatter profile (a) for November 13, 2013 at 13:30 CST and (b-c) its corresponding covariance wavelet transform coefficients with increasing magnitudes of 30, 100, and 300m respectively. Wavelet retrieved BLH is shown as the dashed grey line at 750m.




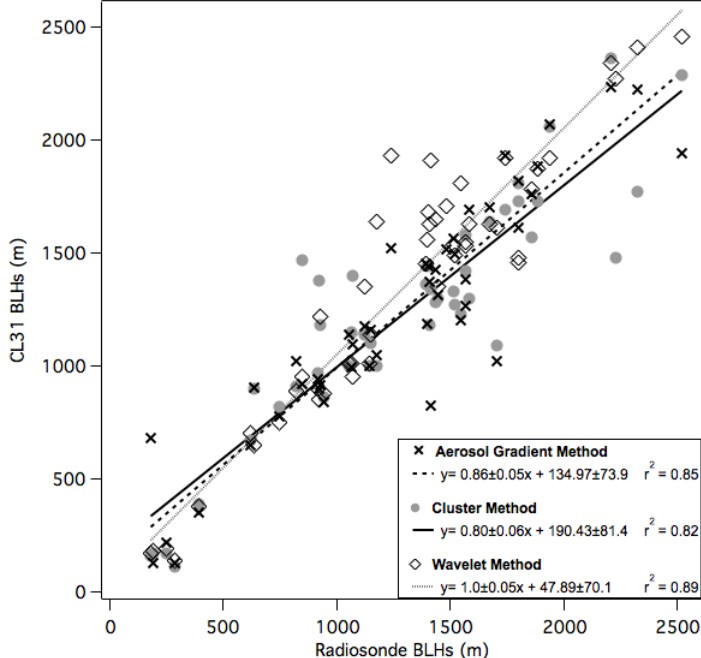

**Figure 5.** Comparison of CL31 aerosol backscatter BLHs and radiosonde derived BLHs. The three methods tested are compared to radiosonde BLHs calculated using the skew-*T log-P* method. The linear regression lines, regression line equations, and correlation coefficients $r^2$ are listed for each BLH retrieval method comparison.




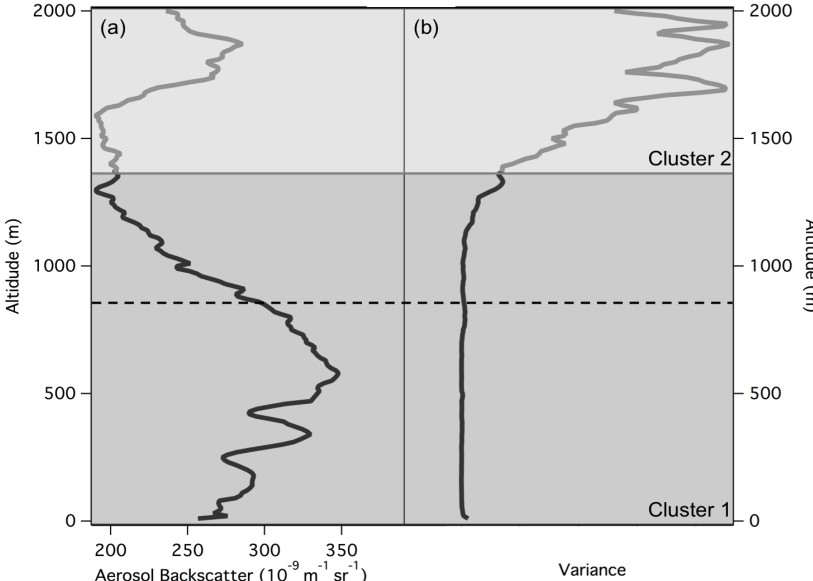

**Figure 6.** Aerosol backscatter profile (a) on October 19, 2013 at 14:00 CST and corresponding calculated variance profile (b) showing division of cluster analysis and estimated BLH (1370m) at the transition from low to high variance. Radiosonde BLH is shown as a dashed line at 850m.





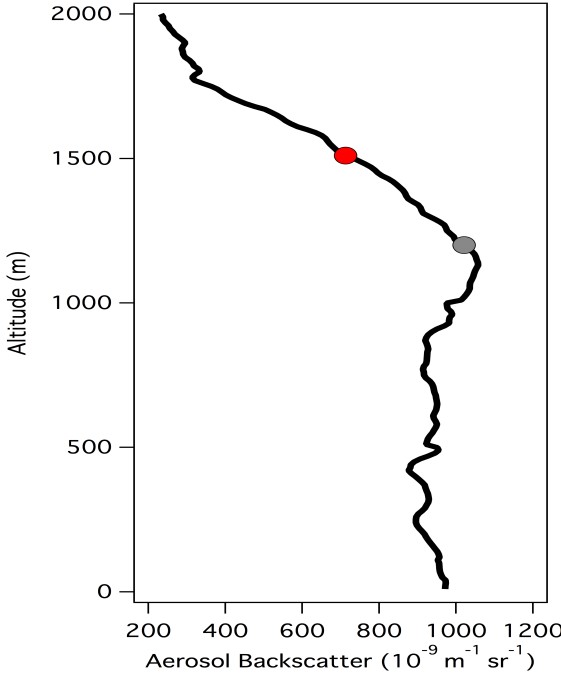

**Figure 7.** Aerosol backscatter profile for October 20, 2014 at 14:00 CST where radiosonde derived BLH does not correspond to the height of the largest negative gradient in the aerosol backscatter profile. Radiosonde BLH at 1290m is shown as a grey circle and wavelet method derived BLH at 1510m is shown as a red circle.





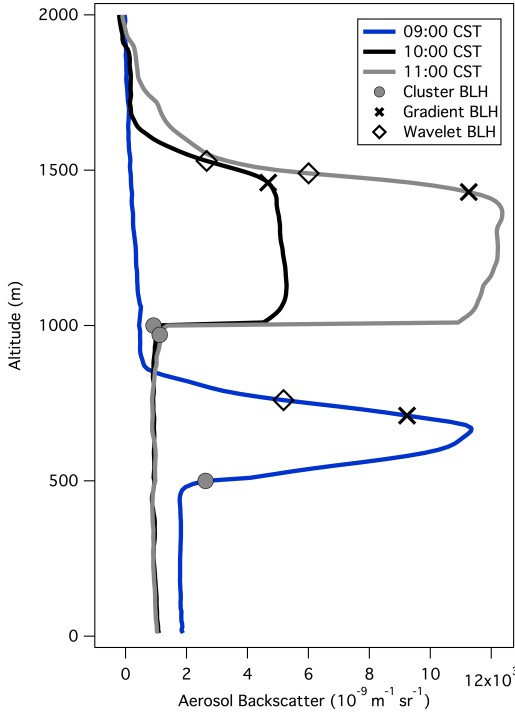

**Figure 8.** Backscatter profiles on September 15, 2013 measured at 09:00 CST (blue), 10:00 CST (black), and 11:00 CST (grey). BLHs retrieved by each method are shown on all profiles. Cloud layers signals measured at about 470-870m, 1000-1620m, and 1000-1520m for 09:00 CST, 10:00 CST and 11:00 CST respectively.




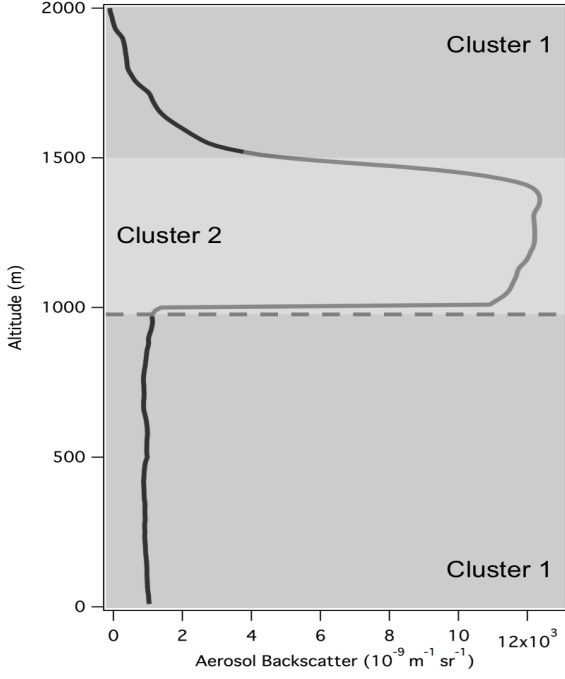

**Figure 9.** Cluster assignments of aerosol backscatter profile with cloud layer at about 1000-1520m on September 15, 2013 measured at 11:00 CST. Automated BLH was found at 970m.