# Peer review of "Comparison of aerosol LIDAR retrieval methods for boundary layer height detection using ceilometer backscatter data"

_Atmospheric Measurement Techniques, 2016_

## Referee Comment (RC1) · Anonymous Referee #1 · 22 Nov 2016

General comments:

In this paper, three algorithms for estimating the boundary layer height (BLH) with a Vaisala CL31 ceilometer are assessed: the manufacturer's aerosol gradient method, the cluster analysis method of Toledo et al. (2014), and the Haar-wavelet method, described e.g. in Cohn and Angevine (2000) or Davis et al. (2000), as well as in many publications later on. The ceilometer is located near the urban area of Houston, Texas, and the ceilometer BLH retrievals are compared with BLH retrievals from collocated radiosonde measurements. In total, 48 radiosonde launches over more than 4 years are considered, all during daytime (mostly in the early afternoon) and cloud-free periods.

All methods compare reasonably well with the reference radiosonde BLH retrievals,

which is also what the experienced reader would have expected under these time of the day and weather conditions. Limitations are mostly due to noise in the backscatter profiles, which is increasing with altitude, to low aerosol backscatter signal, to low clouds and to decoupled aerosol layers. In some cases, thermodynamically and aerosol derived boundary layer heights, even handpicked, did not coincide, which can be expected since aerosols are only a proxy for the thermodynamic BLH and as such may not correlate perfectly with it.

This study suggests that for the CL31 ceilometer, an appropriate time and height average should be applied before applying BLH retrieval methods, that the Haar-wavelet method is less sensible to noise and low signal than the other two methods, and is thus the most appropriate method out of the three methods tested here for automatic BLH retrievals.

Since the CL31 is a profiling ceilometer found in several places, and calculating aerosol backscatter gradients and variances is part of several published PBLH retrieval algorithms, this paper delivers valuable information about the robustness and the good performance of the Haar-wavelet (with a high dilation value) for this ceilometer and its sensibility to noise when calculating the variance.

However, the wavelet-algorithm, as implemented in this study, seems not complex or robust enough to be able to deal with cases where the strongest gradient in the aerosol backscatter profile does not correspond to the BLH. Also, no climatology of retrievals is presented in this study, such that it is difficult to prognose a generalization of the performance of the algorithm.

The authors claim that retrievals are very difficult in case of clouds. I am unsure about it being very difficult, and would be interested to see in their response a Figure showing the performance of the algorithms on the cloudy days, which represent more than 40% of the dataset.

A comparison study, not with the same but with other algorithms that I judge being more

robust, using the CL31 ceilometer as well, has been published in 2012 (Haeffelin et al.). Also, in another study (de Haij et al., Proc., 2010) the wavelet method was applied on the older generation Vaisala ceilometer LD40. No really new algorithm is presented here, rather a comparison of three already existing algorithms: two of gradient-type, one of variance-type. An algorithm fitting an idealized backscatter profile to the data, as in Steyn et al. (1999), would have been an interesting third type to compare with.

To my knowledge, it is the first publication in which the cluster analysis method of Toledo et al. (2014) is evaluated on a profiling ceilometer. Also, I acknowledge the effort to choose manually the PBLH in the aerosol gradient method and skew-T log-P methods, and to identify the discrepancy sources and their occurrence statistics.

The paper is written clearly and concise. Some typographical errors and English grammar may be corrected for. I recommend the paper for acceptance with minor corrections.

Specific comments and technical corrections sorted by page and line:

Abstract

p.1, l.2 be more specific: "Over 40 cloud-free daytime radiosonde profiles"

p.1, l.6 replace: "The Haar Wavelet method demonstrated to be the most robust"

p.1, l.7-8 shorten the sentence, i.e. replace "only showing limitations (22.5% of all observations) due to the basic assumptions used to derive BLH from aerosol backscatter concentrations rather than errors with the algorithm itself." with "only showing limitations in 22.5% of all observations."

See my comments later about "not an error with the algorithm itself".

p.1, l.8 "Disagreement between thermodynamically and aerosol derived boundary layer heights"

You claim "Overall good agreement" in p.1, l.3 but disagreement here. Replace with:

"Occasional differences between thermodynamically and aerosol derived boundary layer heights"

1. Introduction

p.2, l.12 ", facilitating the monitoring of the nocturnal stable layer, internal aerosol layers and the nighttime residual layer"

What about the ML?

p.2, l.14 replace: "in the need for determining the most reliable and accurate method"

2. Data an Instrumentation

p.2, l. 25-26 "Over 80 radiosonde launches and CL31 backscatter profiles beginning in January 2011 through March2015 are used."

Why not say 85 (see Sect. 2.2) directly? Also, this gives me a launch every 19 days during this time period. Please me more specific (regular launches (i.e. every 19 days)? concentrated on some periods, some seasons?)

p.2, l.27 replace: "The effect of cloud signals is analyzed separately for each method"

p.2, l.27-28 "this data set includes data from the NASA DISCOVERAQ Texas campaign"

What kind of data? Radiosonde measurements? Please specify.

2.1 Vaisala CL31

p.3, l.4 Please be more precise: "The backscatter coefficient \beta or the backscattering cross section per unit volume of air is related to the received power with the following formula:"

I prefer the formulation "is related to the received power" instead of "can be calculated from", because calculating \beta from eq (1) is not straightforward. The CL31 gives you a (scaled) attenuated backscatter coefficient (i.e. C_Vaisala \beta \tau^2) in 10^-9

m^-1sr-1 units, with C_Vaisala around 1. I accept that you call this aerosol backscatter or simply backscatter throughout the paper, as long as you remain consistent. Side comment: the liquid cloud calibration method of O'Connor et al. (2004) is a method that is used to calculate C_Vaisala.

p.3, l.6 I am missing the background term (i.e. $P(x,\lambda) = \ldots + B$) in eq. (1) and in its description in lines 7-10. Please add.

p.3, l.15 Please specify which version of the firmware of the CL31 you use and what noise_h2 setting do you use. With the noise_h2 setting turned off, the CL31 backscatter signal is proportional to $P(x)*2400^2$ instead of $P(x)*x^2$ for all ranges>=2400m (in clear sky), because it is assumed that all data further consists of noise. See Kotthaus et al. (2016), Section 3.2 in particular.

This could have an influence on the retrieval of BLHs higher than 2400m.

2.2 iMet RadioSondes

p.3, l.19 "Radiosonde had a response time of 1s."

Remove, because redundant with p.3, l.20.

p.3, l.22 replace: "and a response time of 2s."

p.3, l.25 "with most launches happening around 13:00 CST"

Please be more specific. Add how many launches happened e.g. between 12:30 and 13:30 (or some other period that you define around 13:00 CST), and how many after that midday period.

3. Boundary Layer Height Retrieval Methods

p.3, l.28 remove "therefore a constant concentration of backscatter"

Indeed, "concentration of backscatter" does not make sense. Also, the growth of aerosols due to swelling does not change their concentration, but changes the

backscattered signal (i.e. constant concentration does not mean necessarily constant backscatter).

p.4, l.1-2 "within 10 minutes before or after radiosonde launch."

Please be more specific. Do you choose the closest in time in the 20min time-window? the mean aerosol-BLH in the 20min time-window? the closest in altitude in the 20min time-window? Or else?

3.1 Skew-T log-P Diagram for Radiosonde Boundary Layer Heights

p.4, l.5 replace: "an unstable boundary layer is identified by having a dry adiabatic lapse rate greater than "

p.4, l.8 replace: "where the temperature profile"

p.4, l.9 replace: "where relative humidity or dew point temperature profiles sharply decrease as seen in the skew-T log-P diagram in Fig. 1b"

p.4, l.11 be more specific: "when comparing ceilometer and radiosonde derived BLHs (both manually) using "

Did you retrieve the BLH from the skew-T log-P diagram manually for each day? Please specify.

Haman et al. (2012) found with this method better correlation coefficients (0.96/0.91 in unstable/stable conditions) than in this study, but it seems to be the same instrument. They use the Vaisala v3.5 algorithm, whereas you use the v1.3 algorithm. Also, they use some "quality check" criterions (minimum gradient strength, relative backscatter change around the gradient of 15 %, minimum gradient height). Did you use these quality checks as well (especially this relative backscatter change around the gradient of 15 %)?

Why using here what is seems like an earlier version of the algorithm? Do you think these aspects explain the better correlation in Haman et al. (2012)? Or is it due to

aging of the optics? Or is it due to the fact that they had measurements mostly during spring and summer, with supposedly stronger convective activity?

Please comment on this quantitative difference in the paper, in Sect. 4.1 for example.

I actually do not expect a correlation coefficient of much more than 0.9 when comparing aerosol gradient BLH with radiosonde BLH, especially if the ML is not fully developed, since aerosol gradients are only a proxy for the thermodynamic BLH, and are not expected to correlate perfectly with it.

3.2 Vaisala Corporation Aerosol Backscatter Gradient

p.4, l.18 replace: "The temperature correction of -10 is an algorithm setting that adjusts the shape"

p.4, l.24 "Therefore, a manual analysis of the algorithm's three resulting layers (Fig. 2) is required in order to prevent the incorrect identification of other aerosol layers".

Did you do this manual analysis in this study? Please specify, since not entirely clear here.

p.4, l.25 "The algorithm gives three maximum negative gradients every 1-minute, these are averaged to 10 minutes for radiosonde comparison".

How are they averaged? Layer by layer? Or by clustering all points inside the 10 minutes into 3 clusters/layers, and then taking the mean of these clusters? Or else?

3.3 Cluster Analysis

p.4, l.30-31 be more specific "The BLH is typically identified as the (temporal) variance local maximum"

p.5, l.8 replace: "the greater the EZ depth the greater the overestimation of the BLH"

p.5, l.9 punctuation, missing point at the end: "whereas aerosol gradient methods can give multiple results."

Comment: the cluster method gives a unique BLH only if you choose the number of clusters to be k=2.

3.3.1 Data Processing for Cluster Analysis and Application

p.5, l.14 replace "Due to increasing noise in backscatter profiles with height" with "Because the range correction needed to invert Eq. 1 increases noise in backscatter profiles with height, "

p.5, l.17 be more specific: "The moving time average"

p.5, l.21 In Eq. (2), the power 2 should be applied on the brackets, not on the averaged profile.

Also, you already used P as the received power in Eq. (1). Use an another letter for this variable.

p.5, l.22 try to add space and correct: "where P(z,space ti) is the averaged LIDAR backscattered signal at time ti and height z, and P is the averaged profile from"

p.5, l.22-23 "from N number of profiles"

How much is N here? The number of profiles corresponding to 10min? Please specify.

p.5, l.24 "K-means is a data-partitioning algorithm that assigns observations"

Explain that the observations are 3D-points where the first dimension is the range, the second the backscattered signal and the third the variance, and that these 3D-points are standardized (Toledo et al. (2014)).

p.5, l.30 punctuation, missing point at the end: "Step 4. Compute the average of the observations in each cluster to obtain new centroid locations."

p.6, l.3 replace "Cluster analysis will typically divide a well-mixed boundary layer into two clusters, one below a peak in variance corresponding the center of the EZ,"

with

"By choosing k=2, cluster analysis will typically divide a well-mixed boundary layer into two clusters, one below a peak in variance corresponding to the center of the EZ,"

Indeed, the cluster method gives two clusters only if you choose the number of clusters to be k=2.

p.6, l.5 "will cause the cluster analysis to assign clusters using other criteria"

I do not understand what you mean with "using other criteria". Replace with

"will cause the cluster analysis to assign clusters somewhere else"

3.4 Haar Wavelet Method

p.6, l.13 replace: "top of the BL by calculating the wavelet transform"

p.6, l.20 be more specific: "Defining the dilation factor a and the range of centers b of the Haar wavelet function"

p.6, l.21-22 "b ranges from the lowest ceilometer recorded backscatter altitude of 10m to a maximum BLH of 2800m"

How do you treat cases where the dilation is more than two times the altitude? Do you use a smaller dilation? Do you append zeros to your signal at "negative altitudes"? Please specify.

p.6, l.22 "This limit was set as no radiosonde derived BLHs were found above 2800m"

Comment: Here you could also justify this "upper climatological value" from the study made by Haman et al. (2012). It is better to avoid, if possible, setting algorithm parameters based on the reference data you compare with.

p.6, l.26 put in the plural form: "at the heights of the biggest aerosol gradients in the backscatter profile"

Indeed, there may be more than on big aerosol gradient (e.g. top of BL and top of lofted aerosol layer).

p.6, l. 24-26 change the order of the 2 sentences, i.e. change:

"Lower dilation values create numerous CWTC local minimums (Fig. 4b) at heights of smaller aerosol gradients in the measured profiles. Larger values create large local minimums (Fig. 4c and 4d) at the height of the biggest aerosol gradients in the backscatter profile (Fig. 4a)." with

"Larger values create large local minimums (Fig. 4c and 4d) at the heights of the biggest aerosol gradients in the backscatter profile (Fig. 4a). Lower dilation values create numerous CWTC local minimums (Fig. 4b) at heights of also smaller aerosol gradients in the measured profiles."

The reason is that at the location of a strong negative gradient, the CWTC with a small dilation factor will also have a (strong) local minimum there.

This is why, in some studies, for robustness reasons, the local minima over the averaged CWTC profile (averaged over multiple dilations, say here 30m to 300m) are searched for. Indeed, the gradient you seek at the top of the BL, under the assumptions you stated in the beginning of Section 3, has a peak in the Wavelet transform at multiple dilations. See for e.g. Cohn and Angevine (2000) Fig. 2 b).

As you already mentioned, going to high dilations however does not apply when seeking the top of the NSL, where only small dilations should be used.

p.6, l.29-30 be more specific, i.e. replace "A higher value of 300m (Fig. 4d) for the dilation factor a is applied for daytime BLHs to identify the sharp transition between ML and FT"

with

"A higher value of 300m (Fig. 4d) for the dilation factor a is applied for daytime BLHs and the strongest CWTC local minimum is used to identify the sharp transition between ML and FT"

4 Results

p.7, l.7 replace "t test" with "t-test"

p.7, l.10 "where X is the mean of the aerosol BLH samples, \mu is the radiosonde BLHs mean, S is the standard deviation of samples,"

p.7, l.13 replace "p value" with "p-value"

p.7, l.13 replace "or 95% confidence" with "or 5% significance level"

p.7, l.20 replace: "following precipitation or during periods of high wind speeds."

p.8, l.1 "(not statistically significant; p < 0.05)."

Not statistically significant means p > 0.05, please recheck this and all other statements you make about statistically significance.

p.8, l.5-6 replace: "when the algorithm did not find strong enough gradients in the backscatter profile"

p.8, l.8-9 reformulate, i.e. replace

"This is due to the assumption in the methodology of using aerosol gradients to detect the BLHs and thermal parameters to detect radiosonde BLHs."

with

"This is due to the difference of assumptions in the methodologies, using aerosol gradients to detect the BLHs on one side and thermal parameters to detect radiosonde BLHs on the other side."

4.2 Cluster Method Results

p.8, l.12 What about the statistical significance for this method?

p.8, l.13-14 "From the 45 comparisons performed, 13.3% showed the algorithm mistakenly finding maximum variance peaks not corresponding not corresponding to the

BL but to noise or other aerosol layers"

You claim later in this paragraph 16 cases with noise and 5 cases with lofted aerosol layers, which makes (16+5)/45=46.7% and not 13.3%. Please change accordingly.

p.8 l.19, replace "CL31 displays a significant increase in noise with increasing altitude."

Note that all lidars in general, not only the CL31, display an increase in noise with height, due to the range-correction.

p.8, l.21-22 "These were not algorithm errors but instead due to the implicit assumptions in using aerosol backscatter for BLH detection (constant concentration of backscatter within the BL and a negative gradient in backscatter corresponds to the top of the BL)".

Change "constant concentration of backscatter" by "constant backscatter" (see my comment on p.3, l.28)

I am not satisfied with the formulation: "these were not algorithm errors but instead due to the implicit assumptions". I trust that you did not make a programming error. Still, these errors are due to limitations of the cluster algorithm, and not due to violations of your basic assumptions.

Indeed, assume you have constant backscatter within the BL and a negative gradient in backscatter at the top of the BL. Now suppose there is a lofted aerosol layer on top of the BL and that your measured backscatter profile is affected by noise. This does not violate your basic assumptions of constant backscatter in the BL and gradient at the top, but still perturbs the algorithm.

The cluster method works best to retrieve the top of the residual layer or of the fully developed mixed layer, assuming no clouds, high signal-to-noise ratio and no lofted aerosol layers above the BL. The noise aspect in particular, affecting the temporal variance, seems to have been an important issue here, and you could for e.g. say that this issue should be kept in mind and tackled when using the cluster- or other

variance-based algorithms on the CL31 (even if you say that later in the conclusions, p.10, l.22).

4.3 Wavelet Method Results

p.9, l.4 Add: "capturing the BLH. Also, it was less affected by noise than the gradient method or the cluster method."

Remark: These conclusions are not surprising, since the wavelet method with 300m dilation means taking the difference of two 150m-averaged backscatter signals. With this, you increase the total amount of signal considered and decrease the variability. The drawback is that with a big dilation of 300m, you risk to detect the top of a lofted aerosol layer instead of the top of the BL, which happens 12.5% of the cases, as you mentioned.

4.4 BLH Retrieval with cloud signals

p.9, l.6 soften the appreciation: "The identification of the BLH is more difficult in the presence of clouds".

Indeed, you say yourself in p.9, l.13-15 that in case of cloud the gradient and wavelet methods often compare well with the thermodynamic BLH.

p.9, l.6-7 What about precipitation or fog events? Are they included into this category of BLH retrieval with cloud signals? Please specify.

p.9, l.26 "The presence of clouds limits the detection of the BLH for all methods due to either the extinction of aerosol backscatter signals above the cloud, or the presence of low clouds mistakenly identified as the BLH."

In case of cumulus or stratocumulus, even if you have extinction above the cloud, the error you make with your ceilometer BLH detection compared to the thermodynamic BLH is acceptable and not a major limitation. The presence of low clouds mistakenly identified as the BLH is similar to the cases where you have a lofted aerosol layer above

the BL, in the sense that the algorithms get "distracted" by those (strong) gradients and or variances and fail to pick the correct gradient at the top of the BLH. This is an attribution problem (linked to the processing of the measurement), but not a problem of the ceilometer measurements.

p.9, l.27-28 soften the appreciation: "Hence the removal of profiles with cloud signals is preferred for the automatic retrieval of the BLH".

The removal of profiles with cloud signals is per se not needed for the automatic retrieval of the BLH. Clouds introduce simply additional uncertainty. The algorithms you use in this paper do not really tackle the attribution problem: retrieved BLHs are here simply defined as the strongest gradient in the profile, or the first gradient from the ground, or the first separation point between two clusters, etc. There are more robust algorithms that have been developed in the last few years and advances have been made in this attribution problem issue, see my comments below for the "conclusions" section.

p.9, l.29 be more specific: "since the moving time averaging performed before the application"

5 Summary and Conclusions

p.10, l.2-3 be more specific: "were compared to over 40 cloud-free daytime radiosonde profiles"

p.10, l.9 replace: "corresponding to the top of the BL by calculating the wavelet transform"

p.10, l.20 "Profiles were also mistakenly divided into relative variance concentrations rather than a peak in variance,"

I do not quite understand the term "relative variance concentrations". Replace with:

"Profiles were also mistakenly divided due to the increasing noise with height rather

than a peak in variance,"

p.10, l.21-22 "Previous knowledge of the BL will aid in identifying these algorithm errors, but is not necessary to obtain a BLH with this method."

Not very clear. Replace with:

"This method being automatic, previous knowledge of the BL aids in identifying such algorithm errors, but is otherwise not necessary."

p.10, l.24 "without previous knowledge of the BL required to derive at a BLH."

Not very clear. Replace with:

"without previous knowledge of the BL required, as this method is also automatic."

p.11, l.1 replace: "signal and a cluster for cloud-free signal", i.e. signal without ending s, because it is along one backscatter profile.

p.11, l.5 be more specific: "more sensitive due to the moving time averaging applied"

p.11, l.3-5 "Limited detection of the BLH in aerosol profiles with cloud signals is seen for all methods due to either the extinction of aerosol backscatter signals above the cloud, or the presence of low clouds mistakenly identified as the BLH"

You did not really show this "limited detection of the BLH in aerosol profiles with cloud signals". (85-48)/85 = 43.5% of the measurements were not analyzed because of cloud presence, which is not negligible. I would be very interested to see a scatter plot exactly like the one in Figure 5, but for the 37 days with clouds. You said yourself in Section 4.4 that the methods tested here, especially the gradient and the wavelet methods, correlated quite well with the skew-T log-P derived BLH, so why actually exclude them from the analysis?

p.11, l.8-10 "The errors found with this method were due to the basic assumptions used to derive BLH from aerosol backscatter concentrations rather than errors with the

algorithm itself."

Again, as in p.8, l.21-22, I am not satisfied with this formulation. Replace with:

"The errors found with this method were due to lofted aerosol layers, low-level clouds and differences in determining BLHs using aerosols and thermodynamically using radiosondes."

p.11, l.10-13 "For this reason, the use of this automated method is recommended for BLH observations with careful determination of the dilation values needed for individual instrument and locations. It is suggested to employ the wavelet method in future studies, in particular for long-term boundary layer evolution studies and spatial analysis of the BL using multiple LIDAR aerosol backscatter measurements."

More results are needed in order to accept these two phrases, as I will explain below. Please reformulate:

"In order to use this method on other instruments and locations, dilation values should be determined carefully and individually. Out of the three methods tested in this study, it is suggested to employ the wavelet method in future studies, in particular for long-term boundary layer evolution studies and spatial analysis of the BL using multiple LIDAR aerosol backscatter measurements."

What do you mean by "evolution" in "boundary layer evolution studies"? Seasonal diurnal cycle? Annual cycle of the e.g. 13:00 CST value? Please specify.

Here you did not show any "climatological results" (i.e. do we recover a diurnal and annual cycle with this automatic wavelet method and do they compare qualitatively well with the cycles depicted in the Haman et al. (2012) study?) This would be the expected second part of the validation of the algorithm (wavelet method) you preconize in your conclusions, now that you have good comparison results with the radiosonde measurements, and especially if you suggest this algorithm as suitable for PBL studies on multiple LIDARs.

I am not sure (at least I need to see it) that you are able to see the full diurnal cycle of the ML with the wavelet method as implemented here. For example, during the morning hours, by taking the strongest local minimum, you risk to still detect the top of the RL instead of the top of the developing CBL. But for the annual cycle of the CBL at say 13:00 CST (where you could roughly assume that the CBL is fully developed), I could imagine it works reasonably well.

There are also other automatic methods than the wavelet method or the cluster method, or in a more general sense gradient or variance methods. I think in particular of the "fitting an idealized backscatter profile to observed profiles" method of Steyn et al. (1999), which is a publication you cited. In addition, you might be confronted to physical inconsistencies if you only take the largest peaks in gradient or variance (especially during the development of the CBL in the morning, you risk to jump back and forth between the still existing RL and the CBL). More robust algorithms aiming to tackle these issues exist (among others, gradient-variance coupling with temporal height tracking (Martucci et al. (2010), coupling with a BL-model (di Giuseppe et al. (2012), Biavati (dissert., 2014)), coupling with surface measurements (Pal et al. (2013)), coupling with graph theory (de Bruine et al. (AMTD, 2016))). Such a more robust method has not been investigated or commented here, and your suggestion to use the Haar-wavelet method (out of all existing methods) is thus not justified enough.

Note that there is a review study of (Haeffelin et al., 2012), where several state-of-the-art algorithms at that time on three different ceilometers, including the CL31, are compared.

Here you did not do a review of all state-of-the-art algorithms, but compared three rather simple methods (one manual, two automatic) with radiosonde measurements, and showed that the wavelet method with a large dilation is a robust method on the CL31 to derive the height of strong gradients in the aerosol backscatter profiles, and that using the temporal variance should be done with precaution on this ceilometer. This is valuable information for future work. You also showed that there are cases

where the aerosol layer based and thermodynamic based PBLHs do not match perfectly, which would be a further example giving input to the discussion on how well both methods compare. To my knowledge, it is also the first time the Cluster method, which is based on a good idea but seems quite sensitive to noise, is evaluated with the CL31 ceilometer, and you showed also that it was a potentially interesting technique to detect cloud layers.

Figure 2

Remove the word "Density" in the z-label "Aerosol Backscatter Density" of the figure, because "backscatter density" does not make sense and it will be more consistent with the "Aerosol Backscatter" that you write in all other plots.

Change the legend: "Aerosol backscatter time series for September 26, 2013."

---

## Referee Comment (RC2) · Anonymous Referee #2 · 27 Nov 2016

General comments:

The subject of this paper is the comparison of three algorithms used for estimation boundary layer height (BLH) from a ceilometer CL31 produced by Vaisala. The comparison is performed with an independent dataset of BLH estimates obtained from co-located radiosonde profiles. The algorithms applied to the ceilometer signals are: the Vaisala Corp. BL Matlab v1.3, a cluster methodology as proposed by Toledo et al. 2014, and a Haar Wavelet method.

The methodology for retrieving BLHs from the ceilometer are described enough, as well as the methodology used for estimating the BLHs from the radiosondes. The results show a good agreement for all the methods considered. However, as also referee 1

suggests this is an obvious result when considering BLHs during daytime in cloud free conditions. The results obtained are a confirmation of those obtained by Haman et al. (2012). Unlike similar works Milroy et al. 2012, Haeffelin et al. 2012, Schäfer 2011, the comparison is performed using only one optical instrument. In their conclusions the authors suggest further studies involving more instruments. However, the authors should include a discussion on how their results can be considered if comparing with other instruments: the CL31 was used in several campaigns together with other ceilometers and lidars.

On my opinion, the most relevant aspect of this paper is the use of the cluster method, which unfortunately seems to be the one performing less well than the other two. The Haar Wavelet method used is the one that performs better. Also this conclusions is perfectly in line with the literature on this topic. In particular the issue of having multiple candidates and the selection methods are explored is a known issue since Endlich 1979 for the gradient method and Davis 2000 for the Haar Wavelet. However, the authors do not face this issue directly, as they use a reference sample, which presents conditions of fully developed boundary layer (13:00 CST).

On Fig. 5 the authors present all the results obtained. However, few things are missing: A cross-method comparison showing the 3 methods agreement with each other. A time series of BLHs estimates, which would be very useful for characterising the site.

It would be useful to know in which season-month there is the highest number of reference BLHs. And more in general, as also referee 1 suggest, a climatology information in this work is missing.

Another aspect stressed in the discussion needs to be considered. The Comparison is performed after filtering the data that exceeds a threshold in in the t-test. However, the way the uncertainties for the retrieved BLHs are estimated. Instead of the standard deviation of a sample of retrieved BLHs, the authors should use a more signal related error, like the one proposed in Biavati et al. 2015. This method could be used also for

estimating the errors on the BLHs retrieved from the skew-T log-P method.

I consider that this work should go through a major revision in order to include: more works where the CL31 was used, a BLH climatology at the site, and a more robust way to assess the uncertainties. On the other hand I agree with the referee 1 and I am not going to repeat the details he already underlined.

Biavati, G., Feist, D. G., Gerbig, C., and Kretschmer, R.: Error estimation for localized signal properties: application to atmospheric mixing height retrievals, Atmos. Meas. Tech., 8, 4215-4230, doi:10.5194/amt-8-4215-2015, 2015

Conor Milroy, Giovanni Martucci, Simone Lolli, Sophie Loaec, Laurent Sauvage, Irène Xueref-Remy, Jošt V. Lavrič, Philippe Ciais, Dietrich G. Feist, Gionata Biavati, and Collin D. O'Down. An Assessment of Pseudo-Operational Ground-Based Light Detection and Ranging Sensors to Determine the Boundary-Layer Structure in the Coastal Atmosphere.Advances in Meteorology, 2012:18, 2012.

M. Haeffelin, F. Angelini, Y. Morille, G. Martucci, S. Frey, G. P. Gobbi, S. Lolli, C. D. O'Dowd, L. Sauvage, I. Xueref-Rémy, B. Wastine, and D. G. Feist. Evaluation of Mixing-Height Retrievals from Automatic Profiling Lidars and Ceilometers in View of Future Integrated Networks in Europe. Boundary-Layer Meteorology, 143(1):49–75, 2012.

K. Schäfer, S. Emeis, M. Höß, R. Friedl, C. Münkel, and P. Suppan. Comparison of continuous detection of mixing layer heights by ceilometer with radiosonde observations. SPIE, 8177:817707–817707–8, 2011. K. J. Davis, N. Gamage, C. R. Hagelberg, C. Kiemle, D. H. Lenschow, and P. P. Sullivan. An Objective Method for Deriving Atmospheric Structure from Airborne Lidar Observations.Journal of Atmospheric and Oceanic Technology, 17(11):1455–1468, Nov 2000.

P. Seibert, F. Beyrich, S.-E. Gryning, S. Joffre, alix Rasmussen, and P. Tercier. Review and intercomparison of operational methods for the determination of the mixing height. Atmospheric Environment, 34:1001–1027(27), 2000.

R. Endlich, F. Ludwig, and E. Uthe. An automatic method for determining the mixing depth from lidar observations.Atmospheric Environment (1967), 13(7):1051–1056, 1979.

---

## Author Comment (AC1) · 10 Jan 2017

We thank Referee 1 for carefully reading our manuscript and for the suggestions for revising and improving our work. Below we provide the Referee's review (in bold) followed by our response to individual comments. For reference and help to find the modifications we have made, we appended a revised version of the manuscript to our responses.

**The wavelet-algorithm, as implemented in this study, seems not complex or robust enough to be able to deal with cases where the strongest gradient in the aerosol backscatter profile does not correspond to the BLH. Also, no climatology of retrievals is presented in this study, such that it is difficult to prognose a**

**generalization of the performance of the algorithm.** We have made a small addition to show the performance of all methods over a clear day. This has been added as Figure 8 -a time series of a day in which the performance of all algorithms is seen and can be compared to results of two radiosonde. All algorithms are able to capture well the NSL, the growth of the BL and the peak BLH with the cluster method showing the most variability due to the detection of lofted aerosol layer signals incorrectly identified as the BLH. The aerosol gradient method and the wavelet method BLHs are very comparable after the manual selection of the aerosol gradient method BLHs.

**The authors claim that retrievals are very difficult in case of clouds. I am unsure about it being very difficult, and would be interested to see in their response a Figure showing the performance of the algorithms on the cloudy days, which represent more than 40% of the dataset. A comparison study, not with the same but with other algorithms that I judge being more robust, using the CL31 ceilometer as well, has been published in 2012 (Haeffelin et al.).** We have also added Figure 13, which shows the performance of all algorithms with the presence of cloud signals. This figure shows the decrease in correlation due to cloud signals on the overall performance for all algorithms. These results are discussed in Section 4.4.

The Haeffeling et al. (2012) study does an excellent comparison of five BLH algorithms. Most of the methods used in this study are algorithms, all which require careful pre and post manual selection or analysis of data. Our study wanted to arrive at the most automated yet reliable method to apply to long-term backscatter data such as the one available in the Houston site (2009-2015) and other similar data sets, which increasingly become available.

**Also, in another study (de Haij et al., Proc., 2010) the wavelet method was applied on the older generation Vaisala ceilometer LD40.** The de Haij et al. (2010) study uses a very promising threshold method to prevent the mistaken detection of the BLH due to lofted aerosol layers, the residual layer and so forth. However, this threshold or quality index is arbitrary chosen and it is independent of the absolute value of the

aerosol backscatter in each profile used. The authors advise to use this quality index as a first step to check the reliability of the resulting BLHs. They also point out that the quality index has problems with multiple aerosol layers (lofted and/or residual) when combined with cloud layers where this quality index is suppressed and does not prevent the incorrect identification of the BLH, and is not reliable in profiles with low backscatter as the quality index is identified using a well pronounced BL under fully convective conditions, only. We believe that although this method works in some conditions, it does not systematically tackle the issue.

**p.1, l.2 be more specific: "Over 40 cloud-free daytime radiosonde profiles"** Since the three methods have different number of radiosonde profiles used, we do not specify this in the Introduction, instead this is explained in Results Section 4 along with the reasons for the number of radiosonde profiles used.

**p.1, l.6 replace: "The Haar Wavelet method demonstrated to be the most robust"** This has been replaced in p.1, l.6.

**p.1, l.7-8 shorten the sentence, i.e. replace "only showing limitations (22.5% of all observations) due to the basic assumptions used to derive BLH from aerosol backscatter concentrations rather than errors with the algorithm itself." with "only showing limitations in 22.5% of all observations."** This has been edited as suggested in p.1, l.6-7.

**p.1, l.8 "Disagreement between thermodynamically and aerosol derived boundary layer heights" You claim "Overall good agreement" in p.1, l.3 but disagreement here. Replace with: "Occasional differences between thermodynamically and aerosol derived boundary layer heights"** This has been replaced as suggested in p.1, l.7-8.

**p.2, l.12 ", facilitating the monitoring of the nocturnal stable layer, internal aerosol layers and the nighttime residual layer" What about the ML?** Here, we wanted to emphasize the ability of ceilometer to measure the BLH not limited to the ML

but further measuring the NSL, internal aerosol layer and the residual layer which are seldom continuously monitored and studied. We have clarified this statement in p.2, l.11-14.

**p.2, l.14 replace: "in the need for determining the most reliable and accurate method"** This has been replaced as suggested in p.2, l.14-15.

**p.2, l. 25-26 "Over 80 radiosonde launches and CL31 backscatter profiles beginning in January 2011 through March2015 are used." Why not say 85 (see Sect. 2.2) directly?** This has been replaced as suggested in p.2, l.27.

**Also, this gives me a launch every 19 days during this time period. Please me more specific (regular launches (i.e. every 19 days)? concentrated on some periods, some seasons?)** We have added a figure (now Figure 1) to address this question and show the seasonal distribution as well as the time slots of the launches. Further explanation in p.2, l.27 - p.3, l.1 and also in Section 2.2 p.3, l.31 – p.4, l.2 has also been added.

**p.2, l.27 replace: "The effect of cloud signals is analyzed separately for each method"** This has been replaced as suggested in p.3, l.2.

**p.2, l.27-28 "this data set includes data from the NASA DISCOVERAQ Texas campaign" What kind of data? Radiosonde measurements? Please specify.** This has been corrected to specify ceilometer and radiosonde data from the NASA DISCOVER-AQ Texas campaign p.3, l.3-4.

**p.3, l.4 Please be more precise: "The backscatter coefficient $\beta$ or the backscattering cross section per unit volume of air is related to the received power with the following formula:" I prefer the formulation "is related to the received power" instead of "can be calculated from", because calculating $\beta$ from eq (1) is not straightforward. The CL31 gives you a (scaled) attenuated backscatter coefficient (i.e. C Vaisala $\beta$ $\tau 2$) in $10 - 9$ m $- 1$sr-1 units, with C Vaisala around 1.** We

have replaced as suggested in p.3, l.11-12.

**I accept that you call this aerosol backscatter or simply backscatter throughout the paper, as long as you remain consistent. Side comment: the liquid cloud calibration method of O'Connor et al. (2004) is a method that is used to calculate C Vaisala.** We have made this consistent throughout the paper by always using "aerosol backscatter".

**p.3, l.6 I am missing the background term (i.e. $P(x,\lambda) = \ldots + B$) in eq. (1) and in its description in lines 7-10. Please add.** This has been added in p.3, l.13.

**p.3, l.15 Please specify which version of the firmware of the CL31 you use and what noise_h2 setting do you use. With the noise_h2 setting turned off, the CL31 backscatter signal is proportional to $P(x)*2400^2$ instead of $P(x)*x^2$ for all ranges>=2400m (in clear sky), because it is assumed that all data further consists of noise. See Kotthaus et al. (2016), Section 3.2 in particular. This could have an influence on the retrieval of BLHs higher than 2400m.** We use the firmware version 1.7 with noise_h2 setting on. We have added this information on p.3, l.23.

**p.3, l.19 "Radiosonde had a response time of 1s."
Remove, because redundant with p.3, l.20.** This has been removed as suggested in p.3, l.27.

**p.3, l.22 replace: "and a response time of 2s."** This has been edited as suggested in .3, l.29.

**p.3, l.25 "with most launches happening around 13:00 CST" Please be more specific. Add how many launches happened e.g. between 12:30 and 13:30 (or some other period that you define around 13:00 CST), and how many after that midday period.** This has been clarified in p.3, l.32 - p.4, l.2 and new Figure 1 shows the hourly time slots of radiosonde launch times.

**p.3, l.28 remove "therefore a constant concentration of backscatter" Indeed,**

"concentration of backscatter" does not make sense. Also, the growth of aerosols due to swelling does not change their concentration, but changes the backscattered signal (i.e. constant concentration does not mean necessarily constant backscatter). This is a very good point. We have edited as suggested in p.4, l.4-5.

**p.4, l.1-2 "within 10 minutes before or after radiosonde launch." Please be more specific. Do you choose the closest in time in the 20min time-window? the mean aerosol-BLH in the 20min time-window? the closest in altitude in the 20min time-window? Or else?** This has been explained in p.4, l.9.

**p.4, l.5 replace: "an unstable boundary layer is identified by having a dry adiabatic lapse rate greater than "** This has been edited as suggested in p.4, l.12.

**p.4, l.8 replace: "where the temperature profile"** This has been edited as suggested in p.4, l.15.

**p.4, l.9 replace: "where relative humidity or dew point temperature profiles sharply decrease as seen in the skew-T log-P diagram in Fig. 1b"** This has been edited as suggested in p.4, l.16.

**p.4, l.11 be more specific: "when comparing ceilometer and radiosonde derived BLHs (both manually) using " Did you retrieve the BLH from the skew-T log-P diagram manually for each day?** Yes, BLHs were retrieved manually. This has been edited as suggested in p.4, l.19.

**Haman et al. (2012) found with this method better correlation coefficients (0.96/0.91 in unstable/stable conditions) than in this study, but it seems to be the same instrument. They use the Vaisala v3.5 algorithm, whereas you use the v1.3 algorithm.** This was a mistake (BL-Matlab Control v1.3 was mistakenly referred to). Version 3.7 was used for this study. This has been edited in p.4, l.21.

**Also, they use some "quality check" criterions (minimum gradient strength, rela-**

[Figure]

tive backscatter change around the gradient of 15 %, minimum gradient height). **Did you use these quality checks as well (especially this relative backscatter change around the gradient of 15 %)?** Yes we use the same 15% sensitivity option and a 30m minimum gradient height. These settings have been added in p.4, l.29-30.

**Why using here what is seems like an earlier version of the algorithm? Do you think these aspects explain the better correlation in Haman et al. (2012)?** We are sorry for the confusion and have corrected to list the correct algorithm version p.4, l.21. **Or is it due to aging of the optics?** This cannot be ruled out. Haman et al. (2012) used data from 2009 – 2011 and in this study we continued the use of the same instrument for our data set (2011-2015).

**Or is it due to the fact that they had measurements mostly during spring and summer, with supposedly stronger convective activity Please comment on this quantitative difference in the paper, in Sect. 4.1 for example. I actually do not expect a correlation coefficient of much more than 0.9 when comparing aerosol gradient BLH with radiosonde BLH, especially if the ML is not fully developed, since aerosol gradients are only a proxy for the thermodynamic BLH, and are not expected to correlate perfectly with it.** As can be seen in Figure 1our radiosondes were mainly launched in May, June, September, and October, which also have strong convective activity. We believe the difference in correlation could be due to the manual analysis used by Haman et al. (2012) since they do not report a BLH "if the height of the [BL] is not clear" (p.705 in Haman et al. (2012)) rather than always reporting a gradient found by this algorithm (as long as the algorithm is able to calculate a gradient) as we do in this study. Haman et al. (2013) do not specify the selection criteria, therefore we cannot expand on the specifics and the effect the manual selection might be creating. An addition was made in Section 4.1 p.8, l.25-27.

**p.4, l.18 replace: "The temperature correction of -10 is an algorithm setting that adjusts the shape"** This has been edited as suggested in p.4, l.25-26.

**p.4, l.24 "Therefore, a manual analysis of the algorithm's three resulting layers (Fig. 2) is required in order to prevent the incorrect identification of other aerosol layers". Did you do this manual analysis in this study? Please specify, since not entirely clear here.** This has been re-written as "Therefore, a manual analysis of the algorithm's three resulting layers (Fig. 3) is required in order to prevent the incorrect identification of other aerosol layers. The algorithm gives three maximum negative gradients every 1-minute of which one is manually chosen as the BLH. These are then averaged to 10 minutes for radiosonde comparison" in p.5, l.1-3.

**p.4, l.25 "The algorithm gives three maximum negative gradients every 1-minute, these are averaged to 10 minutes for radiosonde comparison". How are they averaged? Layer by layer? Or by clustering all points inside the 10 minutes into 3 clusters/layers, and then taking the mean of these clusters? Or else?** We have clarified this in p.5, l.3-4. The Vaisala BL Algorithm outputs three calculated gradients for each 1minute output as Layer1, Layer2 and Layer3, for instance. However, Layer1, Layer2 and Layer3 are not necessarily a continuous measurement of the same gradient. A gradient at height x found in Layer1 can be found as a gradient in Layer2 in the following 1-minute output. Averaging layer by layer may include averaging from multiple gradients unrelated to each other and averaging all layers by time could also average multiple gradients. Choosing the BLH manually before averaging prevents this happening and leaves only one output for each one 1-minute estimate. These are then averaged to 10 minutes.

**p.4, l.30-31 be more specific "The BLH is typically identified as the (temporal) variance local maximum"** This has been edited as suggested in p.5, l.9.

**p.5, l.8 replace: "the greater the EZ depth the greater the overestimation of the BLH"** This has been edited as suggested in p.5, l.17.

**p.5, l.9 punctuation, missing point at the end: "whereas aerosol gradient methods can give multiple results." Comment: the cluster method gives a unique**

**BLH only if you choose the number of clusters to be k=2.** This has been edited as suggested in p.5, l.18.

**p.5, l.14 replace "Due to increasing noise in backscatter profiles with height" with "Because the range correction needed to invert Eq. 1 increases noise in backscatter profiles with height,"** This has been edited as suggested in p.5, l.23-24.

**p.5, l.17 be more specific: "The moving time average"** This has been edited as suggested in p.5, l.27.

**p.5, l.21 In Eq. (2), the power 2 should be applied on the brackets, not on the averaged profile. Also, you already used P as the received power in Eq. (1). Use an another letter for this variable.** This has been edited as suggested in p.5, l.31.

**p.5, l.22 try to add space and correct: "where P(z,spaceti) is the averaged LIDAR backscattered signal at time ti and height z, and P is the averaged profile from"** This has been edited as suggested in p.6, l.1.

**p.5, l.22-23 "from N number of profiles" How much is N here? The number of profiles corresponding to 10min? Please specify.** This is correct. We have specified this in p.6, l.1-2.

**p.5, l.24 "K-means is a data-partitioning algorithm that assigns observations" Explain that the observations are 3D-points where the first dimension is the range, the second the backscattered signal and the third the variance, and that these 3D-points are standardized (Toledo et al. (2014)).** This has been specified in p.6, l.4.

**p.5, l.30 punctuation, missing point at the end: "Step 4. Compute the average of the observations in each cluster to obtain new centroid locations."** This has been edited as suggested in p.6, l.10.

**p.6, l.3 replace "Cluster analysis will typically divide a well-mixed boundary layer into two clusters, one below a peak in variance corresponding the center of the**

**EZ,"** with **"By choosing k=2, cluster analysis will typically divide a well-mixed boundary layer into two clusters, one below a peak in variance corresponding to the center of the EZ,"** Indeed, the cluster method gives two clusters only if you choose the number of clusters to be k=2. This has been edited as suggested in p.5, l.15-16.

**p.6, l.5 "will cause the cluster analysis to assign clusters using other criteria" I do not understand what you mean with "using other criteria". Replace with "will cause the cluster analysis to assign clusters somewhere else"** We have clarified with "however profiles with increasing noise and/or lofted aerosol layers will cause the cluster analysis to assign clusters elsewhere (for detailed description of criteria see Results Section 4)" p.6, l.16-17.

**p.6, l.13 replace: "top of the BL by calculating the wavelet transform"** This has been edited as suggested in p.6, l. 26.

**p.6, l.20 be more specific: "Defining the dilation factor a and the range of centers b of the Haar wavelet function"** This has been edited as suggested in p.7, l.3.

**p.6, l.21-22 "b ranges from the lowest ceilometer recorded backscatter altitude of 10m to a maximum BLH of 2800m" How do you treat cases where the dilation is more than two times the altitude? Do you use a smaller dilation? Do you append zeros to your signal at "negative altitudes"? Please specify.** We have edited this section as follows in order to further explain and specify the use of smaller dilations in p.6, l.29 – p.7, l.7-20.

**p.6, l.22 "This limit was set as no radiosonde derived BLHs were found above 2800m" Comment: Here you could also justify this "upper climatological value" from the study made by Haman et al. (2012). It is better to avoid, if possible, setting algorithm parameters based on the reference data you compare with.** This is an excellent point. We have used Haman et al. (2012) and Rappenglück et al. (2008) here to also justify this upper limit p.7, l.5-6.

**p.6, l.26 put in the plural form: "at the heights of the biggest aerosol gradients in the backscatter profile" Indeed, there may be more than on big aerosol gradient (e.g. top of BL and top of lofted aerosol layer).** This has been edited as suggested in p.7, l.10.

**p.6, l. 24-26 change the order of the 2 sentences, i.e. change: "Lower dilation values create numerous CWTC local minimums (Fig. 4b) at heights of smaller aerosol gradients in the measured profiles. Larger values create large local minimums (Fig. 4c and 4d) at the height of the biggest aerosol gradients in the backscatter profile (Fig. 4a)." with "Larger values create large local minimums (Fig. 4c and 4d) at the heights of the biggest aerosol gradients in the backscatter profile (Fig. 4a). Lower dilation values create numerous CWTC local minimums (Fig. 4b) at heights of also smaller aerosol gradients in the measured profiles." The reason is that at the location of a strong negative gradient, the CWTC with a small dilation factor will also have a (strong) local minimum there.** This has been edited as suggested along with the corresponding figure p.7, l.8-11.

**This is why, in some studies, for robustness reasons, the local minima over the averaged CWTC profile (averaged over multiple dilations, say here 30m to 300m) are searched for. Indeed, the gradient you seek at the top of the BL, under the assumptions you stated in the beginning of Section 3, has a peak in the Wavelet transform at multiple dilations. See for e.g. Cohn and Angevine (2000) Fig. 2 b). As you already mentioned, going to high dilations however does not apply when seeking the top of the NSL, where only small dilations should be used.** We use the mean of the averaged CWT coefficients such as Cohn and Angevine (2000), Compton et al. (2013), Scarino et al. (2014) as well, and have improved the description of the method in Section 3.4. p.7, l.7-20.

**p.6, l.29-30 be more specific, i.e. replace "A higher value of 300m (Fig. 4d) for the dilution factor a is applied for daytime BLHs to identify the sharp transition between ML and FT" with "A higher value of 300m (Fig. 4d) for the dilution factor**

**a is applied for daytime BLHs and the strongest CWTC local minimum is used to identify the sharp transition between ML and FT"** This has been reworded as suggested and replaced in p.7, l.14-15.

**p.7, l.7 replace "t test" with "t-test"** This has been edited as suggested in p.7, l.25.

**p.7, l.10 "where X is the mean of the aerosol BLH samples, $\mu$ is the radiosonde BLHs mean, S is the standard deviation of samples,"** This has been edited as suggested in p.7, l.28.

**p.7, l.13 replace "p value" with "p-value"** This has been edited as suggested in p.7, l.31.

**p.7, l.13 replace "or 95% confidence" with "or 5% significance level"** This has been edited as suggested in p.7, l.31.

**p.7, l.20 replace: "following precipitation or during periods of high wind speeds."** This has been edited as suggested in p.8, l.15.

**p.8, l.1 "(not statistically significant; $p < 0.05$)." Not statistically significant means $p > 0.05$, please recheck this and all other statements you make about statistically significance.** This has been edited as suggested in p.8, l.30.

**p.8, l.5-6 replace: "when the algorithm did not find strong enough gradients in the backscatter profile"** This has been edited as suggested in p.8, l.34.

**p.8, l.8-9 reformulate, i.e. replace "This is due to the assumption in the methodology of using aerosol gradients to detect the BLHs and thermal parameters to detect radiosonde BLHs." with "This is due to the difference of assumptions in the methodologies, using aerosol gradients to detect the BLHs on one side and thermal parameters to detect radiosonde BLHs on the other side."** This has been replaced as follows "This is due to the different assumptions in the methodologies when using aerosol gradients to detect LIDAR BLHs or thermal parameters to detect radiosonde BLHs" p.9, l.3-4.

**p.8, l.12 What about the statistical significance for this method?** An additional column in Table 2 has been created showing p-values for all methods.

**p.8, l.13-14 "From the 45 comparisons performed, 13.3% showed the algorithm mistakenly finding maximum variance peaks not corresponding not corresponding to the BL but to noise or other aerosol layers" You claim later in this paragraph 16 cases with noise and 5 cases with lofted aerosol layers, which makes (16+5)/45=46.7% and not 13.3%. Please change accordingly.** We have specified as to what the percentages correspond to and have clarified these in p.9, l.23-30. 16 cases (35.5%) where the cluster analysis divided a cluster by noise i.e. smaller variance in lower altitudes and higher variance in higher altitudes. The 13.3% corresponds to instances were a single clear peak in variance (such as that in the ideal case in Fig. 4) was calculated but this peak corresponded to either noise or lofted aerosol layers (5 lofted aerosol layer cases plus 1 case where noise created one long peak in variance).

**p.8 l.19, replace "CL31 displays a significant increase in noise with increasing altitude." Note that all lidars in general, not only the CL31, display an increase in noise with height, due to the range-correction.** This has been edited as suggested in p.9, l.28.

**p.8, l.21-22 "These were not algorithm errors but instead due to the implicit assumptions in using aerosol backscatter for BLH detection (constant concentration of backscatter within the BL and a negative gradient in backscatter corresponds to the top of the BL)". Change "constant concentration of backscatter" by "constant backscatter" (see my comment on p.3, l.28)** This has been edited as suggested in p.9, 1.23-33.

**I am not satisfied with the formulation: "these were not algorithm errors but instead due to the implicit assumptions". I trust that you did not make a programming error. Still, these errors are due to limitations of the cluster algorithm, and not due to violations of your basic assumptions. Indeed, assume you have**

**constant backscatter within the BL and a negative gradient in backscatter at the top of the BL. Now suppose there is a lofted aerosol layer on top of the BL and that your measured backscatter profile is affected by noise. This does not violate your basic assumptions of constant backscatter in the BL and gradient at the top, but still perturbs the algorithm.** This is correct, cluster analysis noise errors such as the example in Figure 9 do not violate our basic assumptions when we are dealing with uncertainties introduced by noise. These are errors from the "limitations of the cluster algorithm" (not programing errors), which we call "algorithm errors". Errors made by the algorithms which violate basic assumptions of aerosol backscatter algorithms such as BLHs identified from lofted aerosol layers or the residual layers, are considered separately since these assumption errors will occur across all aerosol backscatter algorithms. These would not be called algorithm errors but rather errors arising from our assumptions. Here we sought to differentiate and separate errors in this cluster analysis algorithm from those independent errors not seen in the other two algorithms (the noise errors).

The example of a noise affected lofted aerosol layer could certainly be an error due to the cluster analysis noise sensitivity or the basic assumptions we use. We identify the assumption or algorithm errors based on how the BLH was identified. Therefore, if the profile is affected by noise the identification of the BLH could happened in the separation of clusters into similar variance intensity as seen for 35.5% of cases referred to as "algorithm errors" or the BLH could be identified as the height of the lofted aerosol layer. The later BLH identification case would not be called an algorithm error but rather an error due to the assumptions used.

We have edited this sentence to clarify the distinction in p.9, l.23-33.

**The cluster method works best to retrieve the top of the residual layer or of the fully developed mixed layer, assuming no clouds, high signal-to-noise ratio and no lofted aerosol layers above the BL. The noise aspect in particular, affecting the temporal variance, seems to have been an important issue here, and you**

**could for e.g. say that this issue should be kept in mind and tackled when using the clusteror other variance-based algorithms on the CL31 (even if you say that later in the conclusions, p.10, l.22).** A similar statement has been added to p.12, l.26-27.

**p.9, l.4 Add: "capturing the BLH. Also, it was less affected by noise than the gradient method or the cluster method." Remark: These conclusions are not surprising, since the wavelet method with 300m dilation means taking the difference of two 150m-averaged backscatter signals. With this, you increase the total amount of signal considered and decrease the variability. The drawback is that with a big dilation of 300m, you risk to detect the top of a lofted aerosol layer instead of the top of the BL, which happens 12.5% of the cases, as you mentioned.** This has been edited as follows "This method was also less affected by noise than the gradient method or the cluster method" p.10, l.22. The selection of dilations is certainly a compromise and one that was shown to work for the majority of cases.

**p.9, l.6 soften the appreciation: "The identification of the BLH is more difficult in the presence of clouds". Indeed, you say yourself in p.9, l.13-15 that in case of cloud the gradient and wavelet methods often compare well with the thermodynamic BLH.** This has been edited as suggested in p.11,l.2.

**p.9, l.6-7 What about precipitation or fog events? Are they included into this category of BLH retrieval with cloud signals? Please specify.** Fog and precipitation events are not included in this category. We cannot include these conditions since the comparable radiosonde launches were not performed during fog or precipitation events. This has been specified in p.11, l.5.

**p.9, l.26 "The presence of clouds limits the detection of the BLH for all methods due to either the extinction of aerosol backscatter signals above the cloud, or the presence of low clouds mistakenly identified as the BLH." In case of cumulus or stratocumulus, even if you have extinction above the cloud, the error you**

**make with your ceilometer BLH detection compared to the thermodynamic BLH is acceptable and not a major limitation. The presence of low clouds mistakenly identified as the BLH is similar to the cases where you have a lofted aerosol layer above the BL, in the sense that the algorithms get "distracted" by those (strong) gradients and or variances and fail to pick the correct gradient at the top of the BLH. This is an attribution problem (linked to the processing of the measurement), but not a problem of the ceilometer measurements.** We have replaced this statement with "The presence of clouds creates difficulties in the detection of the BLH for all methods due to the extinction of aerosol backscatter signals above the cloud, the presence of low clouds mistakenly identified as the BLH, or the detection of high cloud signals above the skew-T log-P derived BLH" p.11, l.30. Indeed the cloud problem is similar to lofted aerosol layers where the algorithms are confused by these strong signals. However, the difficulty lies within the difference of low-level clouds, cumulus or stratocumulus clouds were a BL algorithm cannot differentiate between types of clouds, therefore is not able to accept (cumulus/stratocumulus cloud) or reject (low level clouds) BLHs calculated with cloud signals. Additionally, according to literature radiosonde BLH detection methods disagree or sometimes fail in some cloud conditions. For example, Hennemuth and Lammert (2006) show detection methods failing in non-fully convective conditions where the cloud top does not equal the inversion corresponding to the BLH. This case is seen in our data set as well (Figure 13) where radiosonde BLHs show a lower BLH than the aerosol backscatter methods which find the BLH either at the base or top of a high-level cloud. These situations make the BLH detection particularly difficult in the presence of cloud signals.

**p.9, l.27-28 soften the appreciation: "Hence the removal of profiles with cloud signals is preferred for the automatic retrieval of the BLH".** This has been edited as suggested in p.11, l.32-33.

**The removal of profiles with cloud signals is per se not needed for the automatic retrieval of the BLH. Clouds introduce simply additional uncertainty. The**

**algorithms you use in this paper do not really tackle the attribution problem: re-trieved BLHs are here simply defined as the strongest gradient in the profile, or the first gradient from the ground, or the first separation point between two clus-ters, etc. There are more robust algorithms that have been developed in the last few years and advances have been made in this attribution problem issue, see my comments below for the "conclusions" section.** For this reason we feel cloud signals should be removed when using these specific algorithms in order to get the most reliable BLH detection. Backscatter data alone cannot tell us how to calculate the BLH when clouds are present. Figure 13 shows the effect of these cloud signals on the overall correlation. As can be seen here the correlation decreases for all methods, es-pecially the cluster analysis method, which constantly finds the BLH at the cloud base level. This uncertainty also arrives quite often - as mentioned above - when BLHs are lower than that of the cloud layer in not fully developed convective conditions.

**p.9, l.29 be more specific: "since the moving time averaging performed before the application"** This has been edited as suggested in p.11, l.33 - p.12, l.1.

**p.10, l.2-3 be more specific: "were compared to over 40 cloud-free daytime ra-diosonde profiles"** In order to specify the number of radiosondes used we have added the following sentence "This comparison used 47 radiosondes for the aerosol gradient method, 45 for the cluster analysis method, and 48 for the Haar wavelet method due to limitations implicit to each algorithm (see Results Section 4)" p.12, l.6-8.

**p.10, l.9 replace: "corresponding to the top of the BL by calculating the wavelet transform"** This has been edited as suggested in p.12, l.13.

**p.10, l.20 "Profiles were also mistakenly divided into relative variance concentra-tions rather than a peak in variance," I do not quite understand the term "relative variance concentrations". Replace with: "Profiles were also mistakenly divided due to the increasing noise with height rather than a peak in variance,"** This has been edited as suggested in p.12, l.24-25.

**p.10, l.21-22 "Previous knowledge of the BL will aid in identifying these algorithm errors, but is not necessary to obtain a BLH with this method." Not very clear. Replace with: "This method being automatic, previous knowledge of the BL aids in identifying such algorithm errors, but is otherwise not necessary."** This has been edited as follows "With this automated cluster analysis method, a previous knowledge of the BL aids in identifying such algorithm errors, but is otherwise not necessary" p.12, l.25-26.

**p.10, l.24 "without previous knowledge of the BL required to derive at a BLH." Not very clear. Replace with: "without previous knowledge of the BL required, as this method is also automatic."** This has been edited as suggested in p.13, l.2.

**p.11, l.1 replace: "signal and a cluster for cloud-free signal", i.e. signal without ending s, because it is along one backscatter profile.** This has been edited as suggested in p.13, l.12-13.

**p.11, l.5 be more specific: "more sensitive due to the moving time averaging applied"** This has been edited as suggested in p.13, l.16.

**p.11, l.3-5 "Limited detection of the BLH in aerosol profiles with cloud signals is seen for all methods due to either the extinction of aerosol backscatter signals above the cloud, or the presence of low clouds mistakenly identified as the BLH" You did not really show this "limited detection of the BLH in aerosol profiles with cloud signals". (85-48)/85 = 43.5% of the measurements were not analyzed because of cloud presence, which is not negligible. I would be very interested to see a scatter plot exactly like the one in Figure 5, but for the 37 days with clouds. You said yourself in Section 4.4 that the methods tested here, especially the gradient and the wavelet methods, correlated quite well with the skew-T log-P derived BLH, so why actually exclude them from the analysis?** This statement corresponds to the correlation using cloud-free BLHs (Figure 3). We have added Figure 13 to show the effect of clouds in the overall correlation if these were

not removed from the analysis. As can be seen, correlation decreases for all methods when comparing to skew-T log-P derived BLHs. The gradient methods performs reasonably well in fully convective cloud-topped BL were the difference between the BLH by radiosondes is close to the aerosol derived gradients. However a fully convective cloud-topped BL is not always present and radiosondes will show strong inversions at a lower height than the cloud-top as the gradient methods find. In fully cloud topped BLs, the gradient methods find the BLH at similar heights though usually lightly higher than radiosonde BLHs. The cluster method however finds the cloud base as the BLH significantly underestimating the BLH. Correlation coefficients when using only cloud signals is poor for all methods. We calculated an r2 of 0.36, 0.14, and 0.33 for the aerosol gradient, cluster analysis, and wavelet methods respectively (not shown).

**p.11, l.8-10 "The errors found with this method were due to the basic assumptions used to derive BLH from aerosol backscatter concentrations rather than errors with the algorithm itself." Again, as in p.8, l.21-22, I am not satisfied with this formulation. Replace with: "The errors found with this method were due to lofted aerosol layers, low-level clouds and differences in determining BLHs using aerosols and thermodynamically using radiosondes."** This has been edited as suggested in p.13, l.22-24.

**p.11, l.10-13 "For this reason, the use of this automated method is recommended for BLH observations with careful determination of the dilation values needed for individual instrument and locations. It is suggested to employ the wavelet method in future studies, in particular for long-term boundary layer evolution studies and spatial analysis of the BL using multiple LIDAR aerosol backscatter measurements." More results are needed in order to accept these two phrases, as I will explain below. Please reformulate: "In order to use this method on other instruments and locations, dilation values should be determined carefully and individually. Out of the three methods tested in this study, it is suggested to employ the wavelet method in future studies, in particular for long-term bound-**

**ary layer evolution studies and spatial analysis of the BL using multiple LIDAR aerosol backscatter measurements."** This has been edited as suggested in p.13, l.24-27.

**What do you mean by "evolution" in "boundary layer evolution studies"? Seasonal diurnal cycle? Annual cycle of the e.g. 13:00 CST value? Please specify.** This had been indicated in p.13, l.26.

**Here you did not show any "climatological results" (i.e. do we recover a diurnal and annual cycle with this automatic wavelet method and do they compare qualitatively well with the cycles depicted in the Haman et al. (2012) study?) This would be the expected second part of the validation of the algorithm (wavelet method) you preconize in your conclusions, now that you have good comparison results with the radiosonde measurements, and especially if you suggest this algorithm as suitable for PBL studies on multiple LIDARs.** We now show the algorithms for a full day (October 24th, 2013) as a case study to demonstrate the continuous detection efficiency of these algorithms. This day was chosen as it is one of the few days with two radiosonde launches, which qualify for the cloud-free signal analysis at the same time. The result of all algorithms is shown in the added Figure 8 and discussion in Results Section 4 – 4.3.

**I am not sure (at least I need to see it) that you are able to see the full diurnal cycle of the ML with the wavelet method as implemented here. For example, during the morning hours, by taking the strongest local minimum, you risk to still detect the top of the RL instead of the top of the developing CBL. But for the annual cycle of the CBL at say 13:00 CST (where you could roughly assume that the CBL is fully developed), I could imagine it works reasonably well.** This is correct. The algorithms will most likely detect the top of the RL and for this reason we have added a simple height detection limit of 500m during nighttime hours continuing to two hours after sunrise for which resulting BLHs are shown in Figure 8 and detailed in p.8, l.5-9.

**There are also other automatic methods than the wavelet method or the cluster method, or in a more general sense gradient or variance methods. I think in particular of the "fitting an idealized backscatter profile to observed profiles" method of Steyn et al. (1999), which is a publication you cited.** We did not review the idealized backscatter method since we felt it was less reliable and flexible than the method we chose. Studies by Eresmaa et al. (2006), Munkel et al. (2007) and Munoz and Undurraga (2010), show the idealized method to work well under unstable conditions but less so during stable conditions where this method will often mistakenly choose the RL as the BLH. Munoz and Undurraga showed that the use of the idealized method had trouble with a non-idealized BL such as that in the study of the complex Santiago de Chile basin. We felt the idealized method to be less malleable to a stratified BL often found in nocturnal conditions.

**In addition, you might be confronted to physical inconsistencies if you only take the largest peaks in gradient or variance (especially during the development of the CBL in the morning, you risk to jump back and forth between the still existing RL and the CBL). More robust algorithms aiming to tackle these issues exist (among others, gradient-variance coupling with temporal height tracking (Martucci et al. (2010), coupling with a BL-model (di Giuseppe et al. (2012), Biavati (dissert., 2014)), coupling with surface measurements (Pal et al. (2013)), coupling with graph theory (de Bruine et al. (AMTD, 2016))). Such a more robust method has not been investigated or commented here, and your suggestion to use the Haar-wavelet method (out of all existing methods) is thus not justified enough. Note that there is a review study of (Haeffelin et al., 2012), where several state-of-the-art algorithms at that time on three different ceilometers, including the CL31, are compared. Here you did not do a review of all state-of-the-art algorithms, but compared three rather simple methods (one manual, two automatic) with radiosonde measurements, and showed that the wavelet method with a large dilation is a robust method on the CL31 to derive the height of strong gradients in the aerosol backscatter profiles, and that using the temporal variance should be**

**done with precaution on this ceilometer. This is valuable information for future work.** This is correct, we did not review all algorithms available for aerosol backscatter BL detection, but reviewed what we felt are the most used retrieval methods. The purpose of this study was to arrive at the most reliable and (hopefully) automated algorithm to use when dealing with extensive data sets as is the need when using long-term data sets such are those of the CL31. Although methods exist which attempt to tackle the known problems of residual layers, multiple aerosol layers, and clouds they still very much rely on the manual inspection of all data or extensive supplemental measurements. For example, such as the temporal-height-tracking method as that proposed by Martucci et al. (2010) requires manual identification of the BLH from the two local minima detected. The method proposed by de Haij (2006) requires almost a threshold value to prevent the detection of incorrect BLHs yet can fail in various conditions as this threshold is calculated from a pronounced convective BL. The method using surface measurements (Pal et al., 2013) show great correlation with radiosondes when using micrometeorological measurements to aid in the detection of the BLH using the three gradients calculated from the STRAT2D method. However most CL31 measurements sites are not coupled with extensive micrometeorological measurements. The use of a BL-model by di Giuseppe et al. (2012) shows great improvement, especially during nighttime hours when the RL signals are often incorrectly identified. Yet this can also be corrected as seen in Figure 8 by limiting the detection height limits of both wavelet and cluster methods. In contrast, the coupling with graph theory (de Bruine et al. 2016) method is shown to decrease errors coming from other aerosol layers and decreases the jumps in BLHs implicit for example, to STRAT2D calculation of multiple gradients. This method showed very promising results and which could potentially be applied to a long-term automatic detection of the BLH. Overall, the existing methods are aiding in the correct identification of the BLH. However they do not completely tackle the problem, again due to some limitations inherent to BLH detection based on aerosol backscatter methods. We feel that, although the methods reviewed in this study are simple, equally simple fixes, such as the height detection limit during nighttime hours

and the removal of cloud signals tending to elude the algorithms, give reliable detection of the BLH, in a suitable simple automated manner. This is in particular helpful in air quality studies, which mainly occur in less cloudy conditions.

**You also showed that there are cases where the aerosol layer based and thermodynamic based PBLHs do not match perfectly, which would be a further example giving input to the discussion on how well both methods compare. To my knowledge, it is also the first time the Cluster method, which is based on a good idea but seems quite sensitive to noise, is evaluated with the CL31 ceilometer, and you showed also that it was a potentially interesting technique to detect cloud layers.** This was very interesting to us as well. For future studies we would like to look into this further. The detection of clouds layers with this method coupled with the detection of cloud-free BLH might give us a more robust data set for evaluating BL evolution such as seasonal and diurnal BL evolution and how this compares to the presence of cloud layers.

**Figure 2:Remove the word "Density" in the z-label "Aerosol Backscatter Density" of the figure, because "backscatter density" does not make sense and it will be more consistent with the "Aerosol Backscatter" that you write in all other plots. Change the legend: "Aerosol backscatter time series for September 26, 2013."** This has been edited as suggested to Figure 3. The image was also changed to keep consistency with Figure 8 of the day of October 24, 2013.

Please also note the supplement to this comment:
http://www.atmos-meas-tech-discuss.net/amt-2016-340/amt-2016-340-AC1-supplement.pdf

**Supplement:**

**Comparison of aerosol LIDAR retrieval methods for boundary layer height detection using ceilometer aerosol backscatter data**

Vanessa Caicedo[1], Bernhard Rappenglück[1], Barry Lefer[2], Gary Morris[3], Daniel Toledo[4], and Ruben Delgado[5]

[revised manuscript text omitted]

In order to evaluate the retrieval of BLHs from aerosol LIDARS, we tested three distinct methods. Previous studies have evaluated retrieval methods such as the study done by Haeffelin et al. (2012) reviewing various methods (automated and manual) across three LIDARs. This study in turn, evaluates a gradient method, a Haar Wavelet method, and a Cluster Analysis method to retrieve BLHs using aerosol backscatter measured by a Vaisala CL31 ceilometer located in an urban environment.

20 These BLHs are then compared to radiosonde derived BLHs for validation in order to arrive at the automated algorithm with the least manual inspection required. The effect of cloud signals on the BLH retrieval is also observed in all retrieval methods tested and discussed in this study.

**2   Data and Instrumentation**

This study uses Vaisala CL31 ceilometer data and radiosonde profiles measured at the University of Houston (UH) Main

25 Campus. UH Main Campus is located about $70\,\mathrm{km}$ northwest of the Gulf of Mexico and $5\,\mathrm{km}$ southeast of downtown Houston. The UH CL31 was mounted a top a trailer approximately $3.5\,\mathrm{m}$ above ground and radiosonde launches were performed next to the CL31 trailer. A total of 85 radiosonde profiles from the Tropospheric Ozone Pollution Project were analyzed for this study but only profiles corresponding to cloud-free aerosol backscatter vertical profiles are used for the BLH detection comparison. The Tropospheric Ozone Pollution Project seeks to understand the combination of pre and post frontal conditions ideal to high

30 ozone events in the Houston area using ozonesonde and radiosonde profiles. The project is focused in the Fall and Spring seasons when high ozone events are frequent. This results in the data set used containing $\sim 43\%$ of launches during cloudy pre-frontal conditions with a remaining 48 cloud-free launches in post frontal clear skies. Launches between January 2011 and March 2015 are used with the highest frequency in the months of May, June, September and October. All launches occurred

between 6:00 and 17:00 CST with most radiosondes launching during convective ML hours between 13:00-15:00 CST (Fig. 1). The effect of cloud signals is analyzed separately for each method in Section 4.4. In addition, this data set includes ceilometer and radiosonde data from the NASA DISCOVER-AQ (Deriving Information on Surface conditions from Column and Vertically Resolved Observations Relevant to Air Quality) Texas campaign in September 2013.

**2.1 Vaisala CL31**

The Vaisala CL31 ceilometer operates at a wavelength of 905 nanometers (nm) using an indium gallium arsenide laser diode (InGasAs) system with a 1.2 microjoule (mJ) pulse for 110 nanoseconds (ns) and mean pulse repetition rate of 8192 Hertz (Hz). It uses a single lens design to both transmit and receive light signals. This design reduces the optical crosstalk between transmitter and receiver and in turn increases the signal-to-noise ratio. A beam splitter gives full overlap of the transmitter and receiver field-of-view at an altitude of 70m (Münkel et al., 2007).

The aerosol backscatter coefficient $\beta(x,\lambda)$ or the scattering cross section per unit volume is related to the received power with the following formula:

$$P(x,\lambda) = \frac{c}{2x^2} P_0 A \eta O(x) \Delta t \times \beta(x,\lambda) \tau^2(x,\lambda) + B, \tag{1}$$

where $P$ is the optical power received by the ceilometer from distance $x$, $c$ is the speed of light, $\Delta t$ is the pulse duration, $P_0$ is the average laser power during pulse, $A$ is the area of receiver optics, $\eta$ is the receiver optics' efficiency, $O(x)$ is the range dependent overlap integral between transmitted beam and received, $\tau(x,\lambda)$ is the transmittance of the atmosphere between LIDAR and volume, $\lambda$ is the wavelength of the emitted laser pulse, $x$ is the distance between LIDAR and scattering volume and $B$ is the sum of electronic and optical background noise (Weitkamp, 2005). Aerosol backscatter profiles with signals from clouds, rain, or fog are identified as signals higher than $2000 \times 10^{-9} m^{-1} sr^{-1}$ and were not used for this BLH comparison (Kamp and McKendry, 2010).

The CL31 can measure aerosol backscatter up to 7500m. However, the CL31 does not record these signals, but instead only accumulates aerosol backscatter intensity every 16 seconds with a maximum height of 4500m and 10m resolution. The CL31 ran with firmware v1.7 and noise_h2 on. For more in depth information about the instrument see Münkel et al. (2007).

**2.2 iMet Radiosondes**

Radiosondes launched at UH Main Campus are International Met Systems Incorporated model iMet-1. iMet-1 radiosondes return GPS (Global Positioning System) location, GPS altitude, wind speed and direction, pressure, temperature, and relative humidity with a 1 Hz sampling rate using a 403 MHz transmitter. Radiosondes used here have a resolution of 0.01hPa, a response time of 1s, and an accuracy of 0.5hPa for pressure measurements. Temperature sensing has a resolution of 0.01 °C, accuracy 0.2 °C, and a response time of 2s. The humidity sensors for the radiosondes have a resolution of less than 0.1%, accuracy of 5%, and a response time of 2s. Average ascent rate for all launches was about 5 m/s.

A total of 85 launches were analyzed for this study, but only launches corresponding to cloud-free aerosol backscatter vertical profiles are used in the retrieval method comparison. A resulting 48 launches between March 2012 and March 2015 are used

with only four launches happened before 9:00, six before midday and the remaining 38 launches after midday with the highest number of launches happening from 12:00 - 14:00 CST (see Figure 1).

**3 Boundary Layer Height Retrieval Methods**

All aerosol derived BLH methods presented here are based on two assumptions: 1) the BL contains somewhat constant concentrations of aerosols due to convective and turbulent mixing and 2) the clean FT above will create a negative gradient in aerosol backscatter from higher concentrations within the BL towards lower concentrations in the FT. The local maximum of this gradient is identified as the top of the BL (Steyn et al., 1999). Thermodynamic radiosonde BLHs are calculated using a skew-*T log-P* diagram method and are compared to aerosol derived BLHs calculated from aerosol backscatter profiles closest in time to the radiosonde launch but not exceeding 10 minutes before or after the launch.

**3.1 Skew-*T log-P* Diagram for Radiosonde Boundary Layer Heights**

A stable BL is characterized by having an environmental lapse rate greater than a moist/dry adiabatic lapse rate (Fig. 2a), while an unstable boundary layer is identified by having a dry adiabatic lapse rate greater than the environmental lapse rate (Fig. 2b). Stable profiles BLHs are identified as the top of the shallow stable layer as seen as a strong positive vertical gradient change in temperature and a strong negative gradient in dew point temperature profiles (Fig. 2a). BLHs during unstable conditions are identified as the base of the stable EZ (i.e. temperature inversion) where the temperature profile intersects dry adiabates and/or where relative humidity or dew point temperature profiles sharply decrease as seen in the skew-*T log-P* diagram in Fig. 2b (Stull, 1988; Kovalev and Eichinger, 2004; Haman et al., 2012). A previous study by Haman et al. (2012) found a correlation coefficient of 0.96 during unstable conditions and 0.91 during stable conditions when comparing ceilometer and radiosonde derived BLHs (both manually) using the skew-*T log-P* method.

**3.2 Vaisala Corporation Aerosol Backscatter Gradient**

The Vaisala Corp. BL Matlab *v*3.7 algorithm is used in this study. This algorithm finds negative gradients with increasing altitude in aerosol backscatter profiles following the assumptions discussed in Section 3. A $10 \mathrm{~minute}$ and $120 \mathrm{~meter}$ height averaging is applied to the profile along with a temperature dependence curve of -10 as recommended by Vaisala Corporation (C. Münkel, pers. comm., September 2013) due to the tendency of the CL31 having a curvature in aerosol backscatter profiles with increasing internal temperatures. The temperature correction of -10 is an algorithm setting that adjusts the shape and curve of temperature affected aerosol backscatter profiles with negligible effects on aerosol layer detection (Münkel et al., 2007; Vaisala Oyj, 2011, C. Münkel, pers. comm., April 2016).

The change in aerosol backscatter by height ($d\beta/dx$) is calculated by the algorithm, which then finds the largest three negative gradients with minimum aerosol backscatter of gradient of $200 \times 10^{-9} m^{-1} sr^{-1}$ . This study uses a minimum gradient height setting of 30m along with a sensitivity setting of 15% which requires a 15% change in the relative aerosol backscatter in the vicinity of the possible BLH. The largest of the negative gradients is usually defined as the BL (Münkel et al., 2007; Vaisala

Oyj, 2011) however, the largest negative gradient does not always correspond to the BL (see results Sect. 4). Therefore, a manual analysis of the algorithm's three resulting layers (Fig. 3) is required in order to prevent the incorrect identification of other aerosol layers. The algorithm gives three maximum negative gradients every 1-minute of which one is manually chosen as the BLH. These are then averaged to 10 minutes for radiosonde comparison. The manual approach required to select one of the three maximum negative gradients as the BLH requires a priori knowledge of typical nocturnal and daytime BL heights. In addition, this manual selection analysis can be time-consuming especially when longterm LIDAR data is evaluated.

**3.3 Cluster Analysis**

This method uses variations in the measured aerosol vertical profiles for BLH calculations. The BLH is typically identified as the (temporal) variance local maximum local maximum based on the assumption that the EZ contains high aerosol variability due to clean air masses from the free atmosphere mixing with masses from the BL. The center of the EZ corresponds to the top of the BL (Hooper and Eloranta, 1986; Stull, 1988; Piironen and Eloranta, 1995).

Toledo et al. (2014) tested nonhierarchical and hierarchical cluster analysis on LIDAR retrieved vertical aerosol distribution and its variance. Both cluster methods were found to be reliable in calculating BLHs but with a tendency to overestimate the BLH compared to aerosol backscatter gradient methods. This overestimation was attributed to the gradient methods identifying the BLH as a significant decrease in signal, while the cluster method uses a local maximum in variance corresponding to the middle of the EZ. The maximum negative gradient does not always correspond to the local maximum in variance, in these cases the greater the EZ depth the greater the overestimation of the BLH (Toledo et al., 2014). Nevertheless, the cluster method offers a unique BLH, whereas aerosol gradient methods can give multiple results.

**3.3.1 Data Processing for Cluster Analysis and Application**

Due to low signal-to-noise ratio and noise-generated artifacts, both a 10-minute moving time average and moving height average was applied to raw aerosol backscatter profiles. Height averages were applied as seen in Table 1. These averaging settings were chosen as they created the most reliable cluster calculated BLHs, similar to findings in averaging done for gradient methods (Emeis et al., 2008a, b). Because the range correction needed to invert Eq. 1 increases noise in aerosol backscatter profiles with height, lower averaging was applied to lower altitudes while higher averaging was applied to higher altitudes (Table 1). This study found that these averaging settings worked best on most aerosol profiles and aerosol conditions. Typically, lower averaging than those listed in Table 1 caused artificial variance peaks, while greater averaging smoothed out variance peaks in the aerosol backscatter profiles. The moving time average also leads to more profiles containing cloud signals; therefore only 45 comparisons were found to be valid for this method.

Variance $V$ as a function of height $z$ were then calculated from cloud-free profiles $R$ using the following formula (Hooper and Eloranta, 1986):

$$V(z) = \frac{1}{N-1} \sum_{i=1}^{N} [R(z, t_i) - \bar{R}(z)]^2, \tag{2}$$

where $R(z, t_i)$ is the averaged LIDAR aerosol backscattered signal at time $t_i$ and height $z$, and $\bar{R}$ is the averaged profile from $N$ number of profiles corresponding to 10 minutes.

K-means clustering can then be applied to identify BLHs. K-means is a data-partitioning algorithm that assigns standardized 3-D point observations (height range of profile, aerosol backscatter signal, and variance) to exactly one of k clusters defined by centroids (cluster centers), where k is chosen before the algorithm starts (Anderberg, 1973; Toledo et al., 2014). The algorithm works as follows:

Step 1. Choose k initial cluster centers (centroid).

Step 2. Compute point-to-cluster-centroid Euclidean distances of all observations.

Step 3. Assign each observation to the cluster with the closest centroid.

Step 4. Compute the average of the observations in each cluster to obtain new centroid locations.

Step 5. Repeat steps 2 through 4 until cluster assignments do not change, or the maximum number of iterations is reached, whatsoever occurs first, depending on computational resources (Toledo et al., 2014).

Previous determination of the number of clusters present or needed in the dataset is required for cluster validation, since the number of clusters is a parameter to be introduced into the cluster algorithm (Step 1).

By choosing k=2, cluster analysis will typically divide a well-mixed boundary layer into two clusters, one below a peak in variance corresponding to the center of the EZ, one below a peak in variance corresponding the center of the EZ, and one above the variance peak (Fig. 4), however profiles with increasing noise and/or lofted aerosol layers will cause the cluster analysis to assign clusters elsewhere (for detailed description of criteria see Results Section 4). The maximum height of these clusters are limited by the time of day to prevent the detection of other aerosol layers such as the top of the residual layer during nocturnal hours when only the NSL is of interest. Here, the maximum height for nighttime BL detection is 400m, whereas it is 2800m for daytime BL heights.

**3.4 Haar Wavelet Method**

Aerosol backscatter BLHs are derived with a Covariance Wavelet Transform utilizing the Haar wavelet compound step function with multiple user defined wavelet dilations (Cohn and Angevine, 2000; Davis et al., 2000; Brooks, 2003; Baars et al., 2008; Compton et al., 2013; Uzan et al., 2016). This method identifies the sharp aerosol backscatter gradient corresponding to the top of the BL by calculating the wavelet transform. The Haar wavelet function $h$ is defined as follows:

$$h(\tfrac{z-b}{a}) = \left\{ \begin{array}{ll} -1: & b - \frac{a}{2} \leq z < b \\ +1: & b \leq z \leq b + \frac{a}{2} \\ 0: & elsewhere \end{array} \right\}, \tag{3}$$

where $z$ is the vertical altitude in this application, $a$ is the vertical extent or dilation of the Haar function, and $b$ is the center of the Haar wavelet function. The covariance transform of the Haar wavelet function, $w_f$, is defined as:

$$w_f(a, b) = a^{-1} \int_{z_b}^{z_t} f(z) h(\tfrac{z-b}{a}) dz, \tag{4}$$

where $z_t$ and $z_b$ are the top and bottom altitudes in the aerosol backscatter profile, $f(z)$ is the aerosol backscatter profile as a function of altitude, and $a$ is the normalization factor or the inverse of the dilation, respectively.

Defining the dilation factor $a$ and the range of centers of $b$ of the Haar wavelet function are key in correctly identifying the BLH using aerosol backscatter profiles. In this study, $b$ ranges from the lowest ceilometer recorded aerosol backscatter altitude
5 of 10m to a maximum BLH of 2800m. This limit was set as no previous studies have found BLHs above 2800 and as no radiosonde derived BLHs were found above 2800m (Haman et al., 2012; Rappenglück et al., 2008).

As with previous studies (Brooks, 2003; Baars et al., 2008; Compton et al., 2013; Scarino et al., 2014), the dilation factor $a$ affects the number of covariance wavelet transform coefficients (CWTC) local minimums. Larger values create large local minimums (Fig. 5b and 5c) at the heights of the biggest aerosol gradients in the aerosol backscatter profile (Fig. 5a). Lower
10 dilation values create numerous CWTC local minimums (Fig. 5d) at heights of smaller aerosol gradients in the measured profiles. A range of dilation values is applied to the aerosol backscatter profile. Here we use a maximum dilation of 30m for nighttime BLHs since the NSL tends to have a smaller aerosol backscatter gradient than the above RL creating a need for more than one local minimum (not shown). In these cases, the CWTC local minimum closest to the surface is chosen as the BL. A higher limit of 300m (Fig. 5b) for the dilation factor $a$ is applied for daytime BLHs and the strongest CWTC local minimum is
15 used to identify the sharp transition between ML and FT. This larger dilation value also serves to decrease signals from smaller aerosol gradients below the BLH. Cloud-free CL31 aerosol backscatter profiles are averaged first vertically according to Table 1 followed by a 10-minute average before applying the Haar Wavelet algorithm. The algorithm is applied to each averaged profile with incremental dilations until the maximum dilation factor is reached (30m for nighttime hours and 300m for daytime hours). The mean of all resulting CWT coefficients is then calculated and the local minimum of the mean CWT coefficients is
20 identified as the BLH.

**4   Results**

BLH retrieval methods are evaluated and quantified against radiosonde derived BLHs using bias and standard deviation calculated in accordance to Nielsen-Gammon et al. (2008) and Haman et al. (2012). Here, the bias is the difference between the means of aerosol retrieved BLH and the corresponding radiosonde BLH, and the standard deviation is the root-mean-square
25 value of the departures of the individual pair sample differences from the bias. A two-sided, paired sample t-test is used to define the statistical significance of the bias:

$$t = \frac{\bar{X} - \mu}{S} \sqrt{N}, \tag{5}$$

where $\bar{X}$ is the mean of the aerosol BLH samples, $\mu$ is the radiosonde BLHs mean, $S$ is standard deviation of samples, and $N$ is the number of pair samples.
30 The null hypothesis is defined as unbiased aerosol derived BLHs when compared to radiosonde BLHs. It was not rejected when the calculated t-test value (t) was within $\pm 1.96$ and the p-value was greater than 0.05 or 5% significance level, in alignment with previous approaches (Nielsen-Gammon et al., 2008; Haman et al., 2012). Correlation of all methods to radiosonde

BLHs is shown in Figure 6 and an intercomparison of the methods in Figure 7. The uncertainties from the sensor were not calculated for this study as it the exact aerosol backscatter profiles used in the aerosol gradient method are not given by the Vaisala algorithm and therefore the uncertainties could not be calculated equally across all BLH retrieval methods. However, Biavati et al. (2015) shows a promising new statistical method to review sensor related uncertainties in similar studies.

5     The algorithms were applied to October 24, 2013 when two radiosondes launches corresponded to cloud-free signals. The cluster analysis and wavelet method were subjected to a 500m height detection limit during nighttime BLH detection in order to prevent the detection of RL signals and 2800m two hours after sunrise at 9:30 CST (afternoon BL decoupling not shown). The 500m and 2800m limit is chosen as it is well above the previously identified BLHs in the study area (Haman et al., 2012; Rappenglück et al., 2008). The results are shown in Figure 8.

10   **4.1   Aerosol Backscatter Gradient Method Results**

A previous study done by Haman et al. (2012) found that ceilometer BLHs derived from the aerosol backscatter gradient showed excellent correlation to radiosonde BLHs for both stable and unstable conditions, over a period of two years using more than 60 daytime radiosonde profiles. Haman et al. (2012) found the aerosol backscatter gradient capable of continuously identifying the height of the BL after manually choosing one of the three resulting aerosol layers, with limited detection

15 following precipitation or during periods of high wind speeds. Low aerosol content after rain events through wet deposition of aerosols and dispersion of aerosol due to high winds speeds limit the formation of aerosol layers, therefore limiting the detection of the BLH with aerosol gradients. These limitations however, are less relevant for air quality studies since typically these situations are also accompanied by lower pollutant levels (e.g. through air mass change, enhanced vertical mixing, enhanced dry deposition due to high winds, and wet removal of soluble gases during the preceding precipitation). Late afternoon hours

20 also present a challenge since the discontinuous transition from unstable (ML) to stable boundary layer (NSL) can create multiple aerosol layers (Endlich et al., 1979; Seibert et al., 2000; Haman et al., 2012). This is still an important time period for primary pollutant concentrations as they would still be critically determined by the BLH (in particular during evening rush hour), however the diurnal peak in photochemistry activity for build-up of secondary pollutants has passed making this a less crucial time for these pollutants.

25     This study found similar results using 47 cloud-free radiosondes with a slight difference in correlation most likely be due to the manual analysis used. Haman et al. (2012) does not report a BLH if the height of the BL is not clear while this study always reports a gradient found by the algorithm so long as algorithm is able to calculate a gradient. The manual analysis used in this study resulted in a correlation coefficient ($r^2$) of 0.85 was found (Fig. 6) when comparing the aerosol backscatter gradient BLHs to daytime radiosonde BLHs. A bias of -42.5m and a standard deviations of 209.5m (Table 2) were found

30 (not statistically significant; $p > 0.05$). The bias indicates aerosol gradient method BLHs are generally lower than radiosonde BLHs. The overall agreement shows the ability of this method to calculate the BLH reasonably well once one of the three calculated aerosol backscatter gradients is chosen as the BL. However, this requires *a priori* knowledge of typical BLHs at the measurement site and a manual inspection of aerosol gradients calculated. In addition, limited detection of the BLH was also seen in conditions with low aerosol content when the algorithm did not find strong enough gradients in the aerosol backscatter

profile. No combination of available setting options was found to improve BLH detection in these conditions. Furthermore, disagreement was found when the largest gradient in an aerosol profile does not correspond to the thermodynamic BLH found using radiosonde profiles. This is due to the different assumptions in the methodologies when using aerosol gradients to detect LIDAR BLHs or thermal parameters to detect radiosonde BLHs.

5  Figure 8 shows a time series of BLHs reported after manual analysis and 10 minute averaging of the three calculated aerosol layers (Fig. 3). The gradient method is able to resolve for BLHs under stable and unstable conditions for this October day but underestimates the BLH by about 300m and 170m when compared to the first and second radiosonde launch respectively. Nocturnal BLHs are similar to those calculated by the wavelet and cluster analysis method but occasionally measure a lower NSL than the other two methods, likely due to the difference averages used in the aerosol gradient method. Daytime BLHs

10 after manual selection of the three calculated gradients is seen slightly less variable than those calculated by the cluster analysis and wavelet methods and are ocassionally lower than those calculated by the wavelet method. Overall, all methods are able to capture the NSL, the growth of the BL and the peak BLH reasonably well, with the cluster method showing the most variability due to the detection of lofted aerosol layer signals incorrectly identified as the BLH. The aerosol gradient method and the wavelet method BLHs show very similar results after the manual selection of the aerosol gradient method BLHs.

15 Figure 7 shows the aerosol gradient method having the best correlation with the wavelet method as expected as both search for the maximum aerosol backscatter gradients in a profile, but slightly lower agreement with the variance method. Overall, this method works well under stable and unstable conditions so long as the user is able to identify the correct BLH from the three gradients reported.

**4.2 Cluster Method Results**

20 CL31 BLHs using the cluster method showed a slightly lower correlation than the aerosol gradient method with a correlation coefficient of 0.82 (Fig. 6), a bias of -61.0m and a standard deviation of 243.5m (not statistically significant; see Table 2). Disagreements found between radiosonde and cluster derived BLHs were most commonly due to noise in aerosol backscatter profiles and lofted aerosol layers. From the 45 comparisons performed, 13.3% showed the algorithm finding a maximum variance peak not corresponding to the BL but to noise or other aerosol layers. Sixteen cases (35.5%) were found where

25 noise created multiple variance peaks in higher altitudes therefore the cluster analysis divided aerosol backscatter profiles into clusters of similar variance intensity (Fig. 9) rather than above and below a single variance peak (as seen in Fig. 4). This division underestimated the BLH (bias of -61.0) since the cluster was divided into relatively low variance closer to the surface and high variance in higher altitudes. This is due to the fact that CL31 displays a significant increase in noise with increasing altitude. Five instances were found where the variance maximum did not equal radiosonde derived BLH due to signals from lofted

30 aerosol layers. In these cases a smaller maximum corresponded to the BL. These were not errors due to algorithm limitations created by noise (35.5%) but instead due to the implicit assumptions in using aerosol backscatter for BLH detection (constant aerosol backscatter signals within the BL and a negative gradient in aerosol backscatter corresponds to the top of the BL). When compared to the wavelet and aerosol gradient method, the cluster analysis agrees well with the aerosol gradient method ($r^2$ = 0.82) but lightly less with the wavelet method ($r^2$ = 0.76) as seen in Figure 7.

The errors calculated by other aerosol layers can be seen to occur during October 24, 2013 (Fig 8). Here, the cluster method mistakenly identifies signals higher than the BL some of which the aerosol gradient method also identified (see Fig. 3) but were manually rejected as a possible BLH candidates. When compared to the radiosondes launched in this day the cluster analysis does well but slightly underestimating the BLH by no more than 100m in the first launch and 250m in the second launch. The

5  cluster analysis method does well during the nocturnal hours when the algorithm is limited by height preventing the detection of the RL, but errors occur when the nighttime signals are assigned to clusters according to noise similar to the profile shown in Fig. 9.

**4.3   Wavelet Method Results**

The Haar wavelet method showed excellent agreement when compared to 48 radiosonde BLHs with a correlation coefficient

10  of 0.89 (Fig. 6). Statistical analysis showed a bias of 51.1m (not statistically significant) and a standard deviation of 187.0m (Table 2). Disagreement was found when aerosol backscatter profiles contained multiple sharp gradients corresponding to lofted aerosol layers ($\sim 12.5\%$ of total cases). These shallow aerosol layers often have stronger gradients than that of the BL. In these cases, the second largest gradient is very often the BL ($\sim 67\%$). In addition, another $\sim 10\%$ of total cases showed deviations where the radiosonde derived BLH did not correspond to the greatest gradient in the aerosol profile as shown in

15  Figure 10. This disagreement and positive bias found can be attributed to the differences in determining BLHs using aerosols and thermodynamically using radiosondes. Aerosols can penetrate into the stable layer transporting aerosols to higher altitudes than the BLH (inversion height) causing an overestimation of aerosol derived BLHs (McElroy and Smith, 1991; Seibert et al., 2000). Removing the $\sim 22.5\%$ of deviations falling into the cases described above would improve the correlation drastically ($r^2$ = 0.98). This provides confidence that all potential causes for deviations were identified. Overall, the wavelet method showed

20  the best correlation of all methods when compared to radiosondes. In particular, this method was superior in the detection of BLHs in profiles with low aerosol backscatter. Under these conditions it was able to resolve weaker local maximums thus reasonably capturing the BLH. This method was also less affected by noise than the gradient method or the cluster method.

The wavelet method is shown to perform well with the addition of a height restraint for nocturnal BLH retrievals (Fig. 8) in order to prevent the detection of RL signals or lofted aerosol layers. Other methods to prevent the incorrect detection of the BLH include those proposed by de Haij et al. (2016), Di Giuseppe et al. (2012), and Pal et al. (2013). However, our

25  study uses the height restraint as it has shown to successfully prevent the detection of RL signals in the example shown in Figure 8. Both wavelet estimated BLHs are within 30m of the radiosonde derived BLHs. The comparison with the cluster and gradient methods in Figure 7 shows this method generally agrees well with the aerosol gradient method ($r^2$ = 0.84) but appears to calculate the BLH slightly lower than the wavelet method most likely due to difference averaging quantities used.

30  The correlation with the variance method of $r^2$ = 0.76 is most likely due to the noise sensitivity of the cluster analysis method and the calculation of a BLH by using the variance of an aerosol backscatter profile versus finding a gradient in an aerosol backscatter profile.

**4.4 BLH Retrieval with cloud signals**

The identification of the BLH is more difficult in the presence of clouds when aerosol backscatter algorithms identify the strong signals of the cloud layer as the BLH. Strong cloud signals ($>2000 \times 10^{-9} m^{-1} sr^{-1}$) can limit the detection of the BLH due to the extinction of the aerosol backscatter signals above cloud layers. The effect of these cloud signals is observed for all BLH retrieval methods (not including fog or rain events). Although this study observes daytime cloud signals, continuous ceilometer measurements may find similar signals during nighttime hours therefore our findings are not limited to daytime convective mixed layers.

Figure 11 shows hourly aerosol backscatter profiles for September 15, 2013 and corresponding BLHs retrieved by the aerosol gradient, cluster and wavelet methods. Both aerosol gradient and wavelet methods consistently identify the BLH as the top of the cloud layer due to the large negative gradient created by strong cloud signals. This is often the height of the thermodynamic BL identified using relative humidity and dew point temperature methods, which find the height of the ML as the sharp decrease in moisture at the top of the cloud layer. Low cloud layers however impede the detection of the above BLH therefore the aerosol gradient and wavelet method will mistakenly identify the large gradient of the low cloud layers as the BLH, while the cluster method will identify the BL as the base of the low cloud layer. The aerosol gradient method typically found the BLH at the beginning of the large negative gradient (top of the cloud layer) while the wavelet method calculated the BLH slightly higher than the aerosol gradient method. Differences between these two methods were found to not exceed 200m and could be attributed to the different averaging settings applied for these methods.

The cluster method was found to constantly identify the cloud base as the BLH by assigning aerosol signals into a cluster of cloud signals and a second cluster of cloud-free signals with the first transition (BLH) of these clusters located at the base of the cloud layer (Fig. 12) at 970m. A second transition of clusters is located at the top of the cloud layer (about 1400m) corresponding to the BLHs retrieved by the aerosol gradient and wavelet methods. The cluster method then essentially calculates the cloud layer depth by assigning a cluster solely to the cloud layer.

The effect of clouds in the overall correlation between aerosol backscatter methods and radiosonde BLHs in both cloud and cloud-free profiles is seen in Figure 13. During a fully developed convective cloud topped ML, the aerosol gradient methods agree reasonably well with the radiosonde derived BLHs. However, under less developed MLs the agreement decreases due to the aerosol gradient methods identifying the BLH at the top of a cloud layer while the skew-*T log-P* method finds the BL at a strong inversion lower than the cloud layer. This effect can be seen in the radiosonde BLH range of about 800m) to 1500m in Figure 13. The cluster analysis method showed the highest decrease in correlation due to the detection of the cloud base as the BLH significantly underestimating the BLH.

The presence of clouds creates difficulties in the detection of the BLH for all methods due to the extinction of aerosol backscatter signals above the cloud, the presence of low clouds mistakenly identified as the BLH, or the detection of high cloud signals above the skew-*T log-P* derived BLH. Hence the removal of profiles with cloud signals is preferred for the automatic retrieval of the BLH. This affects the cluster and aerosol gradient methods in particular since the moving time averaging

[revised manuscript text omitted]

The effect of cloud signals in the determination of the BLH showed a clear difference between the negative gradient methods
10  (aerosol backscatter and wavelet methods) and the cluster analysis method. Both aerosol gradient and wavelet methods identify the BLH as the top of the cloud layer where a sharp negative gradient created by strong cloud signals was found, while the cluster method identified the BLH as the base of the cloud layer. The cluster method was found to assign a cluster for cloud signal and a cluster for cloud-free signal along an aerosol backscatter profile (Fig. 12). The automatic detection of the first transition of clusters identifies the BLH as the base of the cloud layer with the second transition at the top of the cloud layer,
15  i.e. it identifies the cloud layer depth. Limited detection of the BLH in aerosol profiles with cloud signals is seen for all methods (Fig. 13) with the cluster and aerosol gradient methods being more sensitive due to the moving time averaging applied expanding cloud signals to a greater number of profiles, consequently eliminating these profiles for BLH detection. Both the wavelet and aerosol aerosol gradient methods agree reasonably well with the radiosonde derived BLHs in a fully developed convective cloud topped ML. Agreement decreases when the aerosol gradient and wavelet methods identify the BLH at the top
20  of a cloud layer while the skew-*T log-P* BLHs are calculated at a height lower than the cloud layer under less developed MLs.

The results presented here demonstrate the ability of the Haar Wavelet method to more accurately detect BLHs than the aerosol gradient and cluster methods while requiring the least amount of manual inspection. The errors found with this method were due to lofted aerosol layers, low-level clouds and differences in determining BLHs using aerosols and thermodynamically using radiosondes. In order to use this method on other instruments and locations, dilation values should be determined
25  carefully and individually. Out of the three methods tested in this study, it is suggested to employ the wavelet method in future studies, in particular for long-term seasonal and diurnal boundary layer studies and spatial analysis of the BL using multiple LIDAR aerosol backscatter measurements. Spatial analysis of the BL can also be done by combining multiple LIDAR aerosol backscatter measurements using the wavelet and cluster analysis methods. These methods were seen to perform well using various LIDAR instruments in studies such as Compton et al. (2013), Scarino et al. (2014), and Toledo et al. (2014). A combi-
30  nation of the wavelet method BLH retrievals during clear skies and the cluster analysis method's ability to calculate cloud depth is also recommended for more robust BL studies to retrieve more information about the boundary layer under both conditions. Although not tested in this study, de Bruine et al. (2016) shows promising results using an automated method which prevents incorrect detection of the BLH using graph theory.

*Acknowledgements.* We wish to thank Christoph Münkel with all assistance provided with the ceilometer and Vaisala BL Matlab algorithm. James Flynn and Sergio Alvarez for assistance in the installation and maintenance of the ceilometer. Part of this work was funded by the Texas Commission of Environmental Quality (TCEQ) and the NASA DISCOVER-AQ project.

**Table 1.** Averaging heights by height range used on aerosol backscatter profiles for cluster and wavelet methods.

| Altitude Range | Averaging Height |
| --- | --- |
| 10-490 m | 70 m |
| 500-990 m | 330 m |
| 1000-1990 m | 590 m |
| 2000-4500 m | 690 m |

**Table 2.** Bias, Standard Deviation, p-value and number of data points (No.) for comparison of BLH retrieval methods to radiosonde BLHs.

| BLH Retrieval Method | Bias (m) | Standard Deviation (m) | p-value | No. |
|---|---|---|---|---|
| Aerosol Gradient | -42.5 | 209.5 | 0.17 | 47 |
| Cluster | -61.0 | 243.5 | 0.10 | 45 |
| Wavelet | 51.1 | 187.0 | 0.07 | 48 |

[Figure]

**Figure 1.** Cloud-free radiosondes used for method comparison specified by the time of launch in CST.

[Figure]

**Figure 2.** Skew-*T log-P* method for BLH detection using temperature (black) and dew point temperature (grey) for (a) stable and (b) unstable conditions with BLH shown as grey dashed line. Soundings from September 26, 2013 at 6:10 CST (a) and May 4, 2014 at 15:40 CST (b).

[Figure]

**Figure 3.** Aerosol backscatter time series for October 24, 2013. Three gradient local minimums are plotted for each 1-minute aerosol backscatter profile.

[Figure]

**Figure 4.** CL31 aerosol backscatter profile (a) and corresponding calculated variance profile (b) for September 25, 2014 at 14:30 CST. Dashed line shows the cluster derived BLH (2360m) at the height where the variance cluster assignment changes from cluster 1 to cluster 2.

[Figure]

**Figure 5.** Daytime aerosol backscatter profile (a) for November 13, 2013 at 13:30 CST and (b-c) its corresponding covariance wavelet transform coefficients with increasing magnitudes of 30, 100, and 300m respectively. Wavelet retrieved BLH is shown as the dashed grey line at 750m.

[Figure]

**Figure 6.** Comparison of CL31 aerosol backscatter BLHs and radiosonde derived BLHs. The three methods tested are compared to radiosonde BLHs calculated using the skew-*T log-P* method. The linear regression lines, regression line equations, and correlation coefficients $r^2$ are listed for each BLH retrieval method comparison.

[Figure]

**Figure 7.** Intercomparison of all methods using cloud-free profiles. One-to-one line in dashed grey and linear regression lines in solid black.

[Figure]

**Figure 8.** Resulting BLH for October 24, 2013 with 10-minute averages for all methods. Radiosonde estimated BLHs are shown as red squares.

[Figure]

**Figure 9.** Aerosol backscatter profile (a) on October 19, 2013 at 14:00 CST and corresponding calculated variance profile (b) showing division of cluster analysis and estimated BLH (1370m) at the transition from low to high variance. Radiosonde BLH is shown as a dashed line at 850m.

[Figure]

**Figure 10.** Aerosol backscatter profile for October 20, 2014 at 14:00 CST where radiosonde derived BLH does not correspond to the height of the largest negative gradient in the aerosol backscatter profile. Radiosonde BLH at 1290m is shown as a grey circle and wavelet method derived BLH at 1510m is shown as a red circle.

[Figure]

**Figure 11.** Aerosol backscatter profiles on September 15, 2013 measured at 09:00 CST (blue), 10:00 CST (black), and 11:00 CST (grey). BLHs retrieved by each method are shown on all profiles. Cloud layers signals measured at about 470-870m, 1000-1620m, and 1000-1520m for 09:00 CST, 10:00 CST and 11:00 CST respectively.

[Figure]

**Figure 12.** Cluster assignments of aerosol backscatter profile with cloud layer at about 1000-1520m on September 15, 2013 measured at 11:00 CST. Automated BLH was found at 970m.

[Figure]

**Figure 13.** Comparison of CL31 aerosol backscatter BLHs and radiosonde derived BLHs including cloud signals. The linear regression lines, regression line equations, and correlation coefficients $r^2$ are listed for each BLH retrieval method comparison.

---

## Author Comment (AC2) · 10 Jan 2017

We thank Referee 2 for carefully reading our manuscript and for the suggestions for revising and improving our work. Below we provide the Referee's review (in bold) followed by our response to individual comments. For reference and help to find the modifications made, we appended a revised version of the manuscript to our responses.

**The subject of this paper is the comparison of three algorithms used for estimation boundary layer height (BLH) from a ceilometer CL31 produced by Vaisala. The comparison is performed with an independent dataset of BLH estimates obtained from colocated radiosonde profiles. The algorithms applied to the ceilometer signals are: the Vaisala Corp. BL Matlab v1.3, a cluster methodology**

[Figure]

**as proposed by Toledo et al. 2014, and a Haar Wavelet method. The methodology for retrieving BLHs from the ceilometer are described enough, as well as the methodology used for estimating the BLHs from the radiosondes. The results show a good agreement for all the methods considered. However, as also referee 1 suggests this is an obvious result when considering BLHs during daytime in cloud free conditions. The results obtained are a confirmation of those obtained by Haman et al. (2012). Unlike similar works Milroy et al. 2012, Haeffelin et al. 2012, Schäfer2011, the comparison is performed using only one optical instrument. In their conclusions the authors suggest further studies involving more instruments. However, the authors should include a discussion on how their results can be considered if comparing with other instruments: the CL31 was used in several campaigns together with other ceilometers and lidars.** We appreciate the reviewer's suggestion. The goal of our study was to arrive at the most reliable automated method testing the same method as Haman et al. (2012) plus the additional two methods we have tested. Our results showed to be the most reliable without using cloud signals (Figure 13). We have added references of some studies that have used our 3 retrieval methods selectively with other instruments in order to evaluate the efficiency of the method on the selected instruments. These references are cited throughout the paper: Toledo et al. (2014) used the cluster analysis on a MicroPulse LIDAR. Compton et al. (2013) applied the wavelet algorithm across 3 instruments: an elastic LIDAR (Elastic Lidar Facility) a MicroPulse LIDAR, and a wind profiler. Lastly, Scarino et al. (2013) used the wavelet method for the NASA High-Spectral Resolution LIDAR. To our knowledge, the gradient method described in our paper, has not been tested on other LIDARs expect for the Vaisala ceilometers (CL31, CL51, LD40 etc.), as it is Vaisala proprietary. These studies tested one of our algorithms across different instruments while we sought to arrive at the most automated method for the retrieval of the BLH using the CL31. We have also added references to the Haeffelin et al. (2012) (p.2, l.16-18), Schäfer et al. (2011) and Milroy et al. (2012) studies (p.2, l.9-11) since they contain valuable information of BLH retrieval methods

using various LIDAR instruments and additional BLH retrieval methods than those used in our study for the CL31 ceilometer.

**On my opinion, the most relevant aspect of this paper is the use of the cluster method, which unfortunately seems to be the one performing less well than the other two. The Haar Wavelet method used is the one that performs better. Also this conclusions is perfectly in line with the literature on this topic. In particular the issue of having multiple candidates and the selection methods are explored is a known issue since Endlich 1979 for the gradient method and Davis 2000 for the HaarWavelet. However, the authors do not face this issue directly, as they use a reference sample, which presents conditions of fully developed boundary layer (13:00 CST).** We have added Figure 1, which shows the seasonal and launch time distribution of the radiosondes used. As shown in Figure 1 launch times were not restricted to 13:00 CST, only. We have also added Figure 8, which displays a time series of all algorithms showing their performance.

**On Fig. 5 the authors present all the results obtained. However, few things are missing: A cross-method comparison showing the 3 methods agreement with each other. A time series of BLHs estimates, which would be very useful for characterising the site.** We have added Figure 7 a method comparison for all three methods with each other and Figure 8 a time series sample of BLHs calculated by each method.

**It would be useful to know in which season-month there is the highest number of reference BLHs. And more in general, as also referee 1 suggest, a climatology information in this work is missing.** We have added both Figures 1 and 8 to further expand on these issues. Figure 1 shows launches used with the highest frequency in the months of May, June, September and October and Figure 8 shows a time series for all methods showing their performance and ability to retrieve the diurnal evolution of the BL. In addition, all our findings are consistent with other climatological studies as those done by Haman et al. (2012) and Rappengluck et al. (2008) in the study area.
**Another aspect stressed in the discussion needs to be considered. The Comparison is performed after filtering the data that exceeds a threshold in in the t-test. However, the way the uncertainties for the retrieved BLHs are estimated. Instead of the standard deviation of a sample of retrieved BLHs, the authors should use a more signal related error, like the one proposed in Biavati et al. 2015. This method could be used also for estimating the errors on the BLHs retrieved from the skew-T log-P method.** The Biavati et al. (2015) offer an excellent method to quantifying sensor uncertainties such as measurements from radiosonde and LIDAR measurements. However, we feel these uncertainties will not aid in the purpose of our study of arriving at the most reliable BLH detection algorithm. As Biavati et al. (2015) state the study's goal was to "estimate a reasonable uncertainty for one singular estimate of MH that depends only on the signals used and their uncertainties" and did not serve to "evaluate the uncertainties that the MH has due to the choice of a method... nor to perform a general study on the best way to estimate mixing height". The uncertainties calculated by Biavati et al.'s approach would not provide us with further information on why the algorithms are correctly or incorrectly identifying the BLHs since our uncertainties largely come from the physics/thermal dynamics in the BLH rather than uncertainties from the sensor. Another point to make is our inability to use this statistical method across all BLH retrieval methods use since the Vaisala aerosol gradient method does not give us the exact individual aerosol backscatter profiles used for the BLH calculation, as it is Vaisala proprietary. The statistical analysis will therefore be incomplete if we cannot evaluate all retrieval methods used in our study equally. For future reference we have cited the statistical method of Biavati et al. (2015) as an excellent method to use in the determination of signal related uncertainties in p.8, l.1-4.

**I consider that this work should go through a major revision in order to include: more works where the CL31 was used, a BLH climatology at the site, and a more robust way to assess the uncertainties. On the other hand I agree with the referee 1 and I am not going to repeat the details he already underlined.** We have elaborated modifications for the issues stated here. Please refer to above comments

and also to the responses made to reviewer 1.

**Biavati, G., Feist, D. G., Gerbig, C., and Kretschmer, R.: Error estimation for localized signal properties: application to atmospheric mixing height retrievals, Atmos. Meas. Tech., 8, 4215-4230, doi:10.5194/amt-8-4215-2015, 2015.** We have added this references as it offers a novel new method to estimate instrumental/sensor uncertainties (p.8, l.1-4).

**Milroy C., Martucci G., Lolli S., Loaec S., Sauvage L., Xueref-Remy I., Lavrič J. V., Ciais P., Feist D. G., Biavati G., and O?Down C. D.: An Assessment of Pseudo-Operational Ground-Based Light Detection and Ranging Sensors to Determine the Boundary-Layer Structure in the Coastal Atmosphere. Adv. Meteor., 2012:18, 2012.** We have added this reference as a sample of using LIDAR instrumentation for BL studies(p.2, l.9-11).

**Haeffelin M., Angelini F., Morille Y., Martucci G., Frey S., Gobbi G. P., Lolli S., O'Dowd C. D., Sauvage L., Xueref-Rémy I., Wastine B., and Feist D. G.: Evaluation of Mixing- Height Retrievals from Automatic Profiling Lidars and Ceilometers in View of Future Integrated Networks in Europe. Bound.-Layer Meteor, 143(1):49?75, 2012.** We have added this study to our references as it also compares different aerosol backscatter algorithm for BLH detection (p.2, l.16-18).

**Schäfer K., Emeis S., Hö$\beta$ M., Friedl R., Münkel C., and Suppan P.: Comparison of continuous detection of mixing layer heights by ceilometer with radiosonde observations. SPIE, 8177:817707?817707?8, 2011.** We have added this reference as a sample of studies usingvarious LIDAR instrumentation for BL studies(p.2, l.9-11).

**Davis, K. J., Gamage, N., Hagelberg, C. R., Kiemle, C., Lenschow, D. H., and Sullivan, P. P.: An objective method for deriving atmospheric structure from airborne lidar observations. J. Atmos. Oceanic Technol., 17, 1455-1468, 2000.** This reference was originally included in our paper. This has been kept in our references.

**Seibert, P., Beyrich, F., Gryning, S. E., Joffre, S., Rasmussen, A., and Tercier, P.: Review and intercomparison of operational methods for the determination of the mixing height. Atmos. Environ., 34, 1001-1027, 2000.** This reference was also originally included in our paper. This has been kept in our references.

**Endlich R., Ludwig F., and Uthe E.: An automatic method for determining the mixing depth from lidar observations. Atmos. Environ., 13(7):1051?1056, 1979.** We have added this study to our references as an important reference which addresses the limitation of the gradient method BLH detection using aerosol backscatter signals p.8, l.21.

Please also note the supplement to this comment:
http://www.atmos-meas-tech-discuss.net/amt-2016-340/amt-2016-340-AC2-supplement.pdf

---

## Author Comment (AC3) · 10 Jan 2017

We appreciate the interest in our manuscript and for referring us toward this recent publication about the calculation of uncertainties from signals such as those of radiosonde and ceilometer measurements. We will include Biavati et al (2015) in the revised manuscript as a general reference to sensor related uncertainties. However, we feel that the uncertainties calculated by the presented method will not provide us with further information for the purpose of our study to arrive at the most reliable method for boundary layer height retrievals using aerosol backscatter data.
* * *